# Complex aneuploidy triggers autophagy and p53-mediated apoptosis and impairs the second lineage segregation in human preimplantation embryos

**Marius Regin[1], Yingnan Lei[1], Edouard Couvreu De Deckersberg[1], Charlotte Janssens[1], Anfien Huyghebaert[1], Yves Guns[2], Pieter Verdyck[1,3], Greta Verheyen[2], Hilde Van de Velde[1,2], Karen Sermon[1†], Claudia Spits[1*†]**

[1]Brussels Health Campus/Faculty of Medicine and Pharmacy, Research Group Genetics Reproduction and Development, Vrije Universiteit Brussel, Brussels, Belgium; [2]Brussels Health Campus, Brussels IVF, Universitair Ziekenhuis Brussel (UZ Brussel), Brussels, Belgium; [3]Brussels Health Campus, Medical Genetics, Universitair Ziekenhuis Brussel (UZ Brussel), Brussels, Belgium

**\*For correspondence:**
Claudia.Spits@vub.be

†These authors contributed equally to this work

**Competing interest:** The authors declare that no competing interests exist.

## eLife Assessment

This study provides **valuable** insights into the cellular responses to complex aneuploidy in human preimplantation embryos. The evidence supporting the claims of the authors is now **convincing** after addressing previous concerns. This work will be of interest to embryologists, geneticists and scholars working on reproductive medicine by increasing our understanding of how human embryos respond to chromosomal abnormalities.

**Abstract** About 70% of human cleavage stage embryos show chromosomal mosaicism, falling to 20% in blastocysts. Chromosomally mosaic human blastocysts can implant and lead to healthy new-borns with normal karyotypes. Studies in mouse embryos and human gastruloids showed that aneuploid cells are eliminated from the epiblast by p53-mediated apoptosis while being tolerated in the trophectoderm. These observations suggest a selective loss of aneuploid cells from human embryos, but the underlying mechanisms are not yet fully understood. Here, we investigated the cellular consequences of aneuploidy in a total of 125 human blastocysts. RNA-sequencing of trophectoderm cells showed activated p53 pathway and apoptosis proportionate to the level of chromosomal imbalance. Immunostaining corroborated that aneuploidy triggers proteotoxic stress, autophagy, p53-signaling, and apoptosis independent from DNA damage. Total cell numbers were lower in aneuploid embryos, due to a decline both in trophectoderm and in epiblast/primitive endoderm cell numbers. While lower cell numbers in trophectoderm may be attributed to apoptosis, aneuploidy impaired the second lineage segregation, particularly primitive endoderm formation. This might be reinforced by retention of NANOG. Our findings might explain why fully aneuploid embryos fail to further develop and we hypothesize that the same mechanisms lead to the removal of aneuploid cells from mosaic embryos.

## Introduction

Aneuploidy is common in human preimplantation embryos from both natural (*Munné et al., 2020a*) and medically assisted reproduction cycles (*Capalbo et al., 2021*; *Chavez et al., 2012*; *Chow et al., 2014*; *Fragouli et al., 2019*; *Johnson et al., 2010*; *Mertzanidou et al., 2013a*; *Mertzanidou et al., 2013b*; *Popovic et al., 2020*; *Starostik et al., 2020*; *Vanneste et al., 2009*; *Voullaire et al., 2000*; *Wells and Delhanty, 2000*; *Yang et al., 2021*). Depending on the study, up to 80% of human cleavage stage embryos are found to be aneuploid, of which 70% show chromosomal mosaicism, i.e., the presence of at least two cell lineages with different genomic content (*Regin et al., 2022*). The reasons behind this high rate of chromosomal abnormalities of mitotic origin is still largely unknown (*McCoy, 2017*). One explanation is that cleavage-stage embryos have a weak spindle assembly checkpoint (SAC) that is uncoupled of apoptosis until the blastocyst stage (*Jacobs et al., 2017*; *Vázquez-Diez et al., 2019*), in combination with an abundancy of transcripts of anti-apoptotic genes (*Yan et al., 2013*). Only after embryonic genome activation, the rapidly increasing expression of pro-apoptotic genes along with the full functionality of the SAC establishes the control of mitotic errors. Additionally, aneuploidy in human embryos originates from an erroneous first mitotic division through a prolonged prometaphase (*Currie et al., 2022*).

It is by now well established that chromosomally mosaic embryos can result in healthy new-borns with normal karyotypes, implying a progressive and selective loss of aneuploid cells during development (*Capalbo et al., 2021*; *Fragouli et al., 2019*; *Fragouli et al., 2017*; *Greco et al., 2015*; *Munné et al., 2017*; *Munné et al., 2020b*; *Viotti et al., 2021*; *Yang et al., 2021*). In the preimplantation embryo, several studies have shown that the proportion of aneuploid cells within the embryo decreases from 3 d post-fertilization (3dpf) (*Fragouli et al., 2019*; *van Echten-Arends et al., 2011*; *Yang et al., 2021*) with no apparent preferential allocation of aneuploid cells to either trophectoderm (TE) or inner cell mass (ICM) (*Capalbo et al., 2013*; *Popovic et al., 2018*; *Ren et al., 2022*; *Starostik et al., 2020*). Conversely, the TE of mouse embryos treated with the SAC inhibitor reversine shows a similar but statistically significant 6% enrichment for aneuploid cells (*Bolton et al., 2016*). Similarly, other studies on human embryos found that the TE contained a slightly higher number of aneuploid cells than the ICM, but without reaching statistical significance (*Griffin et al., 2023*; *Starostik et al., 2020*). Lastly, aneuploid cells have also been found to be excluded as cell debris (*Orvieto et al., 2020*) or allocated to the blastocoel cavity and to peripheral cells that do not participate in the formation of the embryo (*Griffin et al., 2023*).

After implantation, aneuploid cells become progressively more frequent in the trophoblast lineage as compared to the epiblast (EPI) and primitive endoderm (PrE) (*Starostik et al., 2020*). This suggests that the trophoblast cells are more tolerant to aneuploidy, which is in line with the rate of 1–2% of confined placental mosaicism found in the general population (*Kalousek and Vekemans, 1996*). Large copy number variations appear to be mostly confined to placental lineages (*Zamani Esteki et al., 2019*) due to developmental bottlenecks occurring during the first lineage segregation which genetically isolates TE from ICM (*Coorens et al., 2021*). Taken together, there is ample evidence supporting the selective elimination of aneuploid cells from the embryonic lineage during pre- and post-implantation development, while the extraembryonic tissues appear to be more tolerant to genetic imbalances.

From a cellular point of view, aneuploidy has two types of consequences: a generic response and one that depends on the specific aneuploidy. The first type of response is highly conserved and has been found in yeast and plants to mouse and human cells (reviewed in: *Santaguida and Amon, 2015*; *Zhu et al., 2018*; *Chunduri and Storchová, 2019*; *Chunduri et al., 2022*; *Krivega et al., 2022*). Aneuploidy results in gene-dosage defects, leading to an unbalanced protein pool that induces the activation of the mechanisms to restore protein homeostasis, such as the chaperone system, the ubiquitin-proteasome system, and autophagy (*Ariyoshi et al., 2016*; *Dürrbaum et al., 2014*; *Santaguida et al., 2015*; *Stingele et al., 2012*). If these systems are overwhelmed, the accumulation of misfolded or aggregated proteins induces proteotoxic stress which is detrimental to the cells (*Donnelly et al., 2014*; *Huettel et al., 2008*; *Ohashi et al., 2015*; *Oromendia et al., 2012*; *Stingele et al., 2012*; *Tang et al., 2011*; *Torres et al., 2007*; *Williams et al., 2008*). Furthermore, aneuploidy can cause replication stress, which in turn can lead to DNA-damage (*Passerini et al., 2016*; *Santaguida et al., 2017*; *Sheltzer et al., 2011*), p53 activation (*Li et al., 2010*; *Santaguida et al., 2017*; *Thompson and Compton, 2010*) and ultimately to cell cycle arrest and apoptosis (reviewed in: *Santaguida and Amon, 2015*; *Zhu et al., 2018*; *Regin et al., 2022*).

These mechanisms also appear to be active during early embryonic development and to have a lineage-specific effect. In mice, clearance of aneuploid cells by apoptosis requires the activation of autophagy and p53 (*Singla et al., 2020*). Aneuploid cells are only being eliminated by apoptosis in the ICM (*Bolton et al., 2016*) and at similar rates in the EPI and the PrE (*Singla et al., 2020*). In a study using human 2D gastruloids with reversine-induced aneuploidy, apoptosis was only induced in differentiating but not in pluripotent cells, although aneuploidy did induce an increase in total nuclear p53 in both cell types. Similarly, as in the mouse, the aneuploid cells were eliminated by apoptosis from the embryonic germ layers but were tolerated in the TE-like cells (*Yang et al., 2021*).

The type of aneuploidy and the percentage of aneuploid cells within a euploid/aneuploid mosaic embryo determine its developmental capacity. The viability of mouse embryos inversely correlates to the proportion of aneuploid cells, with 50% of euploid cells being sufficient to sustain normal development (*Bolton et al., 2016*). This same threshold appears to apply to human embryos, where embryos with less than 50% of aneuploid cells have higher developmental potential than those containing more than 50%, albeit both performing worse than fully euploid embryos (*Viotti et al., 2021*; *Capalbo et al., 2021*).

Conversely, whole chromosome gains and losses that are present in every cell of the human embryo are usually linked to lethality with the notable exception of trisomies 13 (Patau syndrome), 18 (Edwards syndrome), and 21 (Down syndrome), and sex chromosome abnormalities. The effects of these imbalances can manifest already during early embryo development (*Fuchs Weizman et al., 2019*; *Licciardi et al., 2018*; *Sanchez-Ribas et al., 2019*; *Shahbazi et al., 2020*). Shahbazi et al. recently described the consequences of carrying specific meiotic aneuploidies on human embryo development using an in vitro implantation model (*Shahbazi et al., 2020*). They showed that human embryos with trisomy 15, 16, and 21 and monosomy 21 readily all reached the blastocyst stage, and that monosomy 21 embryos arrested after implantation while trisomy 15 and 21 embryos developed further. Interestingly, trisomy 16 embryos showed a hypoproliferative trophoblast due to excessive CDH1 expression, a gene located on chromosome 16.

In this study, we investigated the cellular responses to complex aneuploidy during human preimplantation development, with a focus on stress response and lineage segregation events. We assessed previously unexplored general consequences of naturally acquired aneuploidies in human embryos with a focus on impaired lineage segregation events.

## Results

### Human embryos with complex aneuploidy show gene dosage defects and transcriptomic signatures of p53 activation and apoptosis

We first tested for the presence of a common transcriptomic response to different complex aneuploidies. For this, we studied the transcriptome of a TE biopsy of fifty human blastocysts (5 or 6dpf) previously diagnosed by Preimplantation Genetic Testing (PGT) as either euploid or as containing an aneuploidy of at least two whole chromosomes in all cases except for one embryo (further referred to as aneuploid, detailed karyotypes can be found in the *Supplementary file 1*). The aneuploidy was presumed to have occurred during meiosis or during the first cleavage from the zygote to the two-cell stage, resulting in all or most of the cells of the TE biopsy containing the same genetic imbalances. We also included a TE sample of eleven human blastocysts that were treated with 0.5 μM reversine from 3dpf to 4dpf, to induce complex mosaic aneuploidy (referred to as reversine-treated embryos). Of these, 14 euploids, 20 aneuploids, and 11 reversine-treated embryos yielded good quality RNA-sequencing results.

Principal component analysis using all expressed genes revealed no clustering of the euploid, aneuploid, or reversine-treated embryos into distinct groups (*Figure 1A*, *Figure 1—figure supplement 1A*). We used InferCNV (https://github.com/broadinstitute/inferCNV/wiki) to bioinformatically infer the chromosomal copy numbers based on the higher or lower expression of the genes located on chromosomes with gains or losses (*Figure 1B*, *Figure 1—figure supplement 1B–C*). We compared the inferred karyotypes to those obtained during PGT and found a match for 45/48 of the full chromosome aneuploidies, while euploidy was correctly predicted in all cases (*Figure 1—figure supplement 1B*). These results show that aneuploidy in human blastocysts results in abnormal gene dosage effects. Of note, the karyotypes of the reversine-treated embryos could not be determined since this

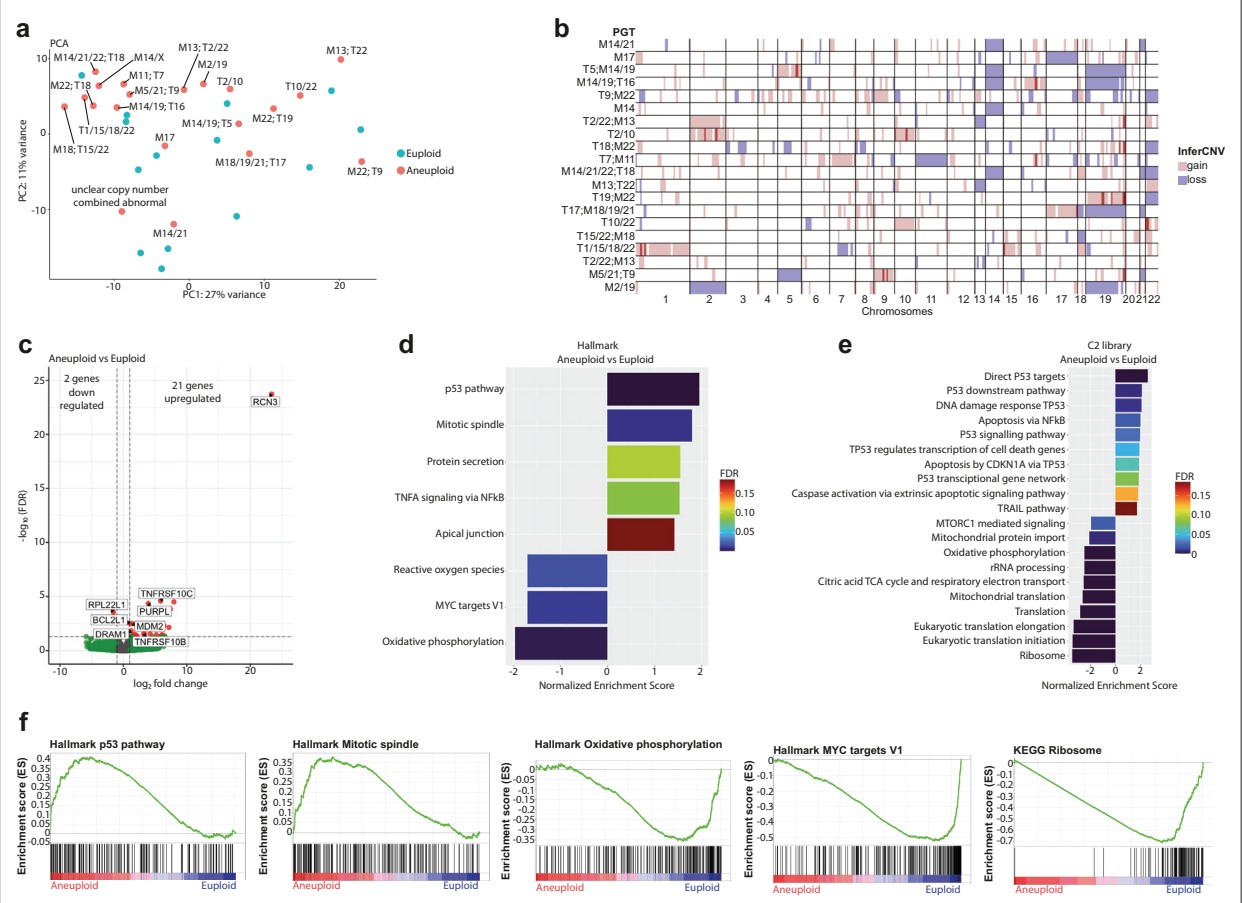

**Figure 1.** RNA-sequencing of trophectoderm cells reveals gene dosage defects and transcriptomic signatures of p53 activation and apoptosis. (**a**) Principal component analysis after transcriptome analysis of aneuploid versus euploid embryos. T=Trisomy, M=Monosomy. (**b**) Comparison between the diagnosis of aneuploid embryos obtained after Preimplantation Genetic Testing (PGT) (list on the left) and the results obtained after using InferCNV after RNA-sequencing (plot on the right). Red and blue indicate gain or loss of the chromosome, respectively. (**c**) Volcano plot after differential gene expression analysis with a cutoff value of |log₂ fold change|>1 and -log₁₀(FDR)<0.05 for aneuploid versus euploid embryos showing 21 upregulated and two downregulated genes. (**d**) Barplot of enriched Hallmark pathways after gene set enrichment analysis of aneuploid versus euploid embryos using a cut-off value of 25% FDR. (**e**) Barplot of TOP20 enriched C2 library pathways after gene set enrichment analysis of aneuploid versus euploid embryos using a cut-off value of 25% FDR. (**f**) Enrichment plots of p53-pathway, mitotic spindle, oxidative phosphorylation, MYC targets V1 and ribosome, showing up- or downregulation of genes that are part of the corresponding pathway. Source of all embryos: Experiment 1.

The online version of this article includes the following source data and figure supplement(s) for figure 1:

**Source data 1.** Count table of RNA-sequencing data.

**Figure supplement 1.** Expanded RNA-sequencing analysis.

drug induces different aneuploidies in each cell, leading to mosaic TE biopsies. Also, the karyotypes of the 3dpf embryos were not determined by PGT before the reversine treatment. InferCNV detected only a trisomy 12 in one of the reversine-treated embryos, likely of meiotic or first-cleavage origin (*Figure 1—figure supplement 1C*).

Despite the abnormal gene dosage, but in line with the lack of clear clustering in the PCA, the differential gene expression analysis yielded 21 significantly upregulated and two downregulated genes (*Figure 1C*, *Supplementary file 2*). These findings are similar to those of other studies on the transcriptome of aneuploid human embryonic cells, which also found a few significantly deregulated genes (*Gallardo et al., 2023*; *Groff et al., 2019*; *Martin et al., 2023*; *Maxwell et al., 2022*). We next focused on pathway analysis based on a ranked gene list of the whole transcriptome. We found that cellular stress gene sets were positively enriched in the aneuploid samples, including the p53-pathway and apoptosis (*Figure 1D–F*). Gene sets related to metabolism, translation, mitochondrial function,

and proliferation, such as translation and ribosome, oxidative phosphorylation and MYC targets, were negatively enriched (*Figure 1D–F*).

When comparing reversine-treated versus euploid embryos we found 34 significantly upregulated and 22 significantly downregulated genes (*Figure 1—figure supplement 1D–F*, *Supplementary file 3*). The expression profile differed from that identified in the endogenously aneuploid embryos. Here, we found mitotic spindle, WNT-beta catenin signaling, KRAS signaling, and lysosome gene sets to be positively enriched while oxidative phosphorylation, MYC targets, translation, and ribosome were negatively enriched. We found no evidence of p53 activation or apoptosis. This suggests that naturally acquired aneuploidies of meiotic origin may elicit a different transcriptomic response than reversine-induced mitotic aneuploidies, with a differential induction of the p53 signaling pathway and apoptosis, and inhibition of proliferation.

## High gene-dosage imbalances result in high p53 activation and apoptosis while low gene dosage imbalances induce pro-survival pathways

To test whether there is a relationship between the type of cellular stress signatures and the size of the chromosomal imbalances, as previously described in cancer cells (*Dürrbaum and Storchová, 2016*), we categorized the aneuploid embryos into a low- and a high- imbalanced gene dosage group. This separation was based on the total number of coding loci that were affected by the aneuploidies (*Supplementary file 4*). The low-dosage group contained the embryos under the 50th percentile when ranking the embryos from the lowest to the highest number of unbalanced loci. The high-dosage group contained the embryos with the 50th percentile or higher. Principal component analysis showed that high-dosage embryos clustered separately from the euploid embryos (*Figure 2A*), with 65 significantly upregulated and 153 significantly downregulated genes (*Figure 2B*, *Supplementary file 5*). High-dosage embryos showed a transcriptomic profile consistent with activation of the p53-pathway and apoptosis, while the unfolded protein response, DNA-repair, MYC targets, oxidative phosphorylation, translation, and ribosome were inhibited (*Figure 2C–E*).

In contrast, low-dosage embryos lacked separation from the euploid embryos in the PCA (*Figure 2F*) and only had one differentially expressed gene (*Figure 2G*). While we still found aneuploidy-related stress responses such as p53-pathway and apoptosis to be activated, we found also pro-survival gene sets such as MYC targets (*Dang et al., 2006*; *Wang et al., 2021*), MTORC1 signaling (*Hung et al., 2012*), unfolded protein response (*Hetz et al., 2020*; *Zanetti et al., 2016*) and hypoxia (*Simões-Sousa et al., 2018*) to be enriched (*Figure 2H–J*). Low-dosage embryos showed no profiles of ribosome or translation impairment. Last, the plotting of the expression levels of genes in the p53 and apoptosis pathways show their progressive stronger induction or suppression from euploid to low- and high-dosage embryos (*Figure 2K*). This illustrates the association between the severity of the protein pool imbalance and the induction of the cellular stress response.

## Aneuploidy triggers proteotoxic stress, DNA-damage independent p53-activation, autophagy, and apoptosis

To further investigate the cellular stress responses identified by the transcriptomic analysis, we carried out immunostaining on the same embryos that underwent TE biopsy for transcriptome analysis. The results show that aneuploid and reversine-treated embryos have increased signal intensity of active CASP3/7 and CASP8 (*Figure 3A–D*). Both aneuploid and reversine-treated embryos also have lower cell numbers than euploid embryos, likely due to the apoptosis-mediated cell loss (*Figure 3E*).

We next sought to corroborate the transcriptomic signatures of proteotoxic stress and autophagy. The number of puncta per cell of LC3B, and p62, as well as the intensity of HSP70 staining, as markers of autophagy and chronic protein misfolding, were significantly increased in both aneuploid and reversine-treated embryos (*Figure 3F–J*). We also tested for the colocalization of CASP8 and HSP70 at the individual cell level. We found that reversine-treated embryonic cells mainly co-express CASP8/HSP70, while the minority is single positive for either CASP8 or HSP70 (*Figure 3K–L*). These results show that although the transcriptomic profile of naturally occurring aneuploidies and reversine-treated embryos is moderately different (*Figure 1C–F*, *Figure 1—figure supplement 1D–F*), the downstream protein response is similar.

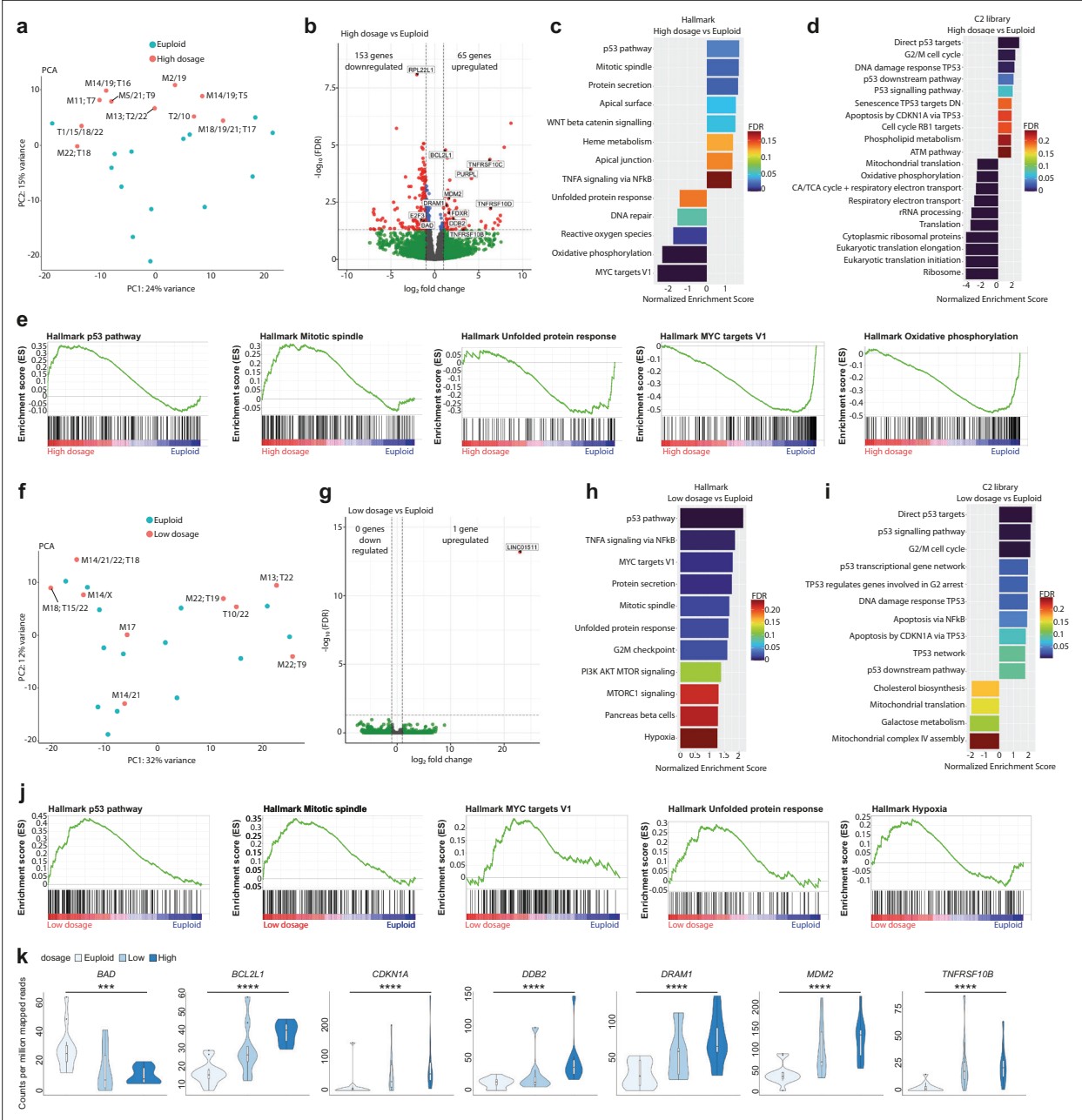

**Figure 2.** Human embryos with the highest number of genes with abnormal copy number show stronger p53 pathway and apoptosis response. (**a**) Principal component analysis after transcriptome analysis of high-dosage aneuploid versus euploid embryos. T=Trisomy, M=Monosomy. (**b**) Volcano plot after differential gene expression analysis with a cutoff value of |log$_2$ fold change|>1 and -log$_{10}$(FDR)<0.05 for high-dosage aneuploid versus euploid embryos showing 65 upregulated and 153 downregulated genes. (**c**) Barplot of enriched Hallmark pathways after gene set enrichment analysis of high-dosage aneuploid versus euploid embryos using a cut-off value of 25% FDR. (**d**) Barplot of TOP20 enriched C2 library pathways after gene set enrichment analysis of high-dosage aneuploid versus euploid embryos using a cut-off value of 25% FDR. (**e**) Enrichment plots of p53-pathway, mitotic spindle, unfolded protein response, MYC targets V1 and oxidative phosphorylation, showing up- or downregulation of genes that are part of the corresponding pathway. (**f**) Principal component analysis after transcriptome analysis of low-dosage aneuploid versus euploid embryos. T=Trisomy, M=Monosomy. (**g**) Volcano plot after differential gene expression analysis with a cutoff value of |log$_2$ fold change|>1 and -log$_{10}$(FDR)<0.05 for low-dosage aneuploid versus euploid embryos showing 1 upregulated and 0 downregulated genes. (**h**) Barplot of enriched Hallmark pathways after gene set enrichment analysis of low-dosage aneuploid versus euploid embryos using a cut-off value of 25% FDR. (**i**) Barplot of enriched C2 library pathways after gene set enrichment analysis of low-dosage aneuploid versus euploid embryos using a cut-off value of 25% FDR. (**j**) Enrichment plots of p53-pathway, mitotic spindle, MYC targets V1, unfolded protein response, and hypoxia showing upregulation of genes that are part of the corresponding pathway. (**k**) Violin plots with box and whisker plots of the counts per million mapped reads of a supervised set of 10 that are part of apoptosis (*BAD, BCL2L1, TNFRSF10B*), p53 pathway (*DRAM1, MDM2, CDKN1A*), and DNA-damage (*DDB2*). *p=0.027, **p=0.010, ***p=0.004, ****p<0.001 using the Jonkheere-Terpstra test. Box and whisker plots show median and minimum to maximum values. Source of all embryos: Experiment 1.

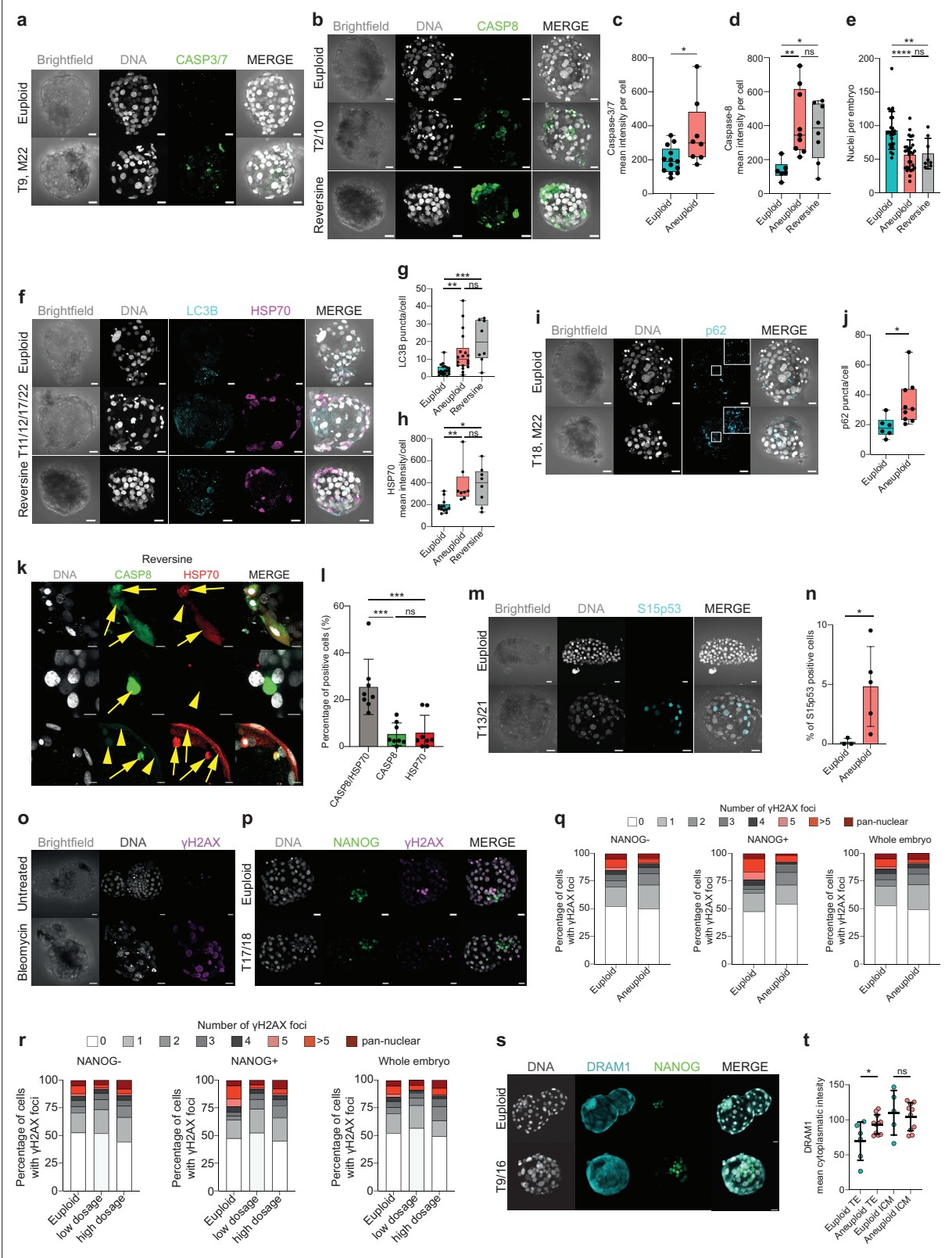

**Figure 3.** Immunostaining reveals increased proteotoxic stress, DNA-damage independent p53-activation, autophagy, and apoptosis. (**a**) Orthogonal projections after immunostaining of euploid and aneuploid embryos for DNA (white) and CASP3/7 (green). T=Trisomy, M=Monosomy. Source: Experiment 1. (**b**) Orthogonal projections after immunostaining of euploid, aneuploid, and reversine-treated embryos for DNA (white) and CASP8 (green). T=Trisomy. Source: Experiment 1. (**c**) CASP3/7 mean intensity per cell. Euploid n=13, Aneuploid n=8 embryos. Unpaired t-test, *p=0.0177.

*Figure 3 continued on next page*

*Figure 3 continued*

Source: Experiment 1. (**d**) CASP8 mean intensity per cell. Euploid n=6, Aneuploid n=9, Reversine-treated n=8 embryos. One-way ANOVA, *p=0.04, **p=0.0098, ns=non-significant. Source: Experiment 1. (**e**) Number of nuclei per embryo. Euploid n=26, Aneuploid n=29, Reversine-treated n=8 embryos. One-way ANOVA, ****p<0.0001, **p=0.0033, ns = non-significant. Source: Experiment 1 and 3. (**f**) Orthogonal projections after immunostaining of euploid, aneuploid, and reversine-treated embryos for DNA (white), LC3B (turquoise), and HSP70 (magenta). T=Trisomy. Source: Experiment 1. (**g**) LC3B puncta per cell. Euploid n=19, Aneuploid n=22 embryos, Reversine-treated n=8 embryos. One-way ANOVA, **p=0.0069, ***p=0.0003, ns=non-significant. Source: Experiment 1. (**h**) HSP70 mean intensity per cell. Euploid n=13, Aneuploid n=8, Reversine-treated n=8 embryos. One-way ANOVA, *p=0.0121, **p=0.0096, ns = non-significant. Source: Experiment 1. (**i**) Orthogonal projections after immunostaining of euploid, aneuploid for DNA (white), and p62 (turquoise). White square shows a zoom of a representative area. T=Trisomy, M=Monosomy. Source: Experiment 1. (**j**) p62 puncta per cell. Euploid n=6, Aneuploid n=9 embryos. Unpaired t-test, *p=0.0284. Source: Experiment 1. (**k**) Optical sections of reversine treated embryos for DNA (white), CASP8 (green), HSP70 (red) showing cells with presence (yellow arrow) or absence (yellow arrowhead) of the proteins to investigate co-localization. Source: Experiment 1. (**l**) Percentage of cells of reversine embryos positive for either CASP8/HSP70, CASP8, or HSP70. One-way ANOVA, CASP8/HSP70 vs CASP8 ***p=0.0003, CASP8/HSP70 vs HSP70 ***p=0.0005, ns = non-significant. Source: Experiment 1. (**m**) Orthogonal projections after immunostaining of euploid, aneuploid for DNA (white), and Serine (**S**) 15 p53 (turquoise). Source: Experiment 4. (**n**) Percentage of Serine 15 p53 positive cells per embryo. Euploid n=3, Aneuploid n=5. Unpaired t-test with Welch's correction, *p=0.0356. T=Trisomy. Source: Experiment 4. (**o**) Orthogonal projections of immunostained untreated (n=6) and Bleomycin treated (n=6) embryos for DNA (white) and gH2AX (magenta) showing few foci in the untreated and pan-nuclear expression of gH2AX. Source: Experiment: 2. (**p**) Orthogonal projections of immunostained euploid (n=7) and aneuploid (n=11) embryos for DNA (white), NANOG (green) and gH2AX (magenta). T=Trisomy. Source: Experiment 3. (**q**) Percentage of cells with gH2AX foci or pan nuclear expression of euploid (n=7) and aneuploid (n=11) embryos in NANOG-negative, NANOG positive cells and whole embryos. Source: Experiment 3. (**r**) Percentage of cells with gH2AX foci or pan nuclear expression of euploid (n=7), low-dosage (n=5), and high-dosage (n=6) aneuploid embryos in NANOG-negative, NANOG positive cells and whole embryos. Source: Experiment 3. (**s**) Orthogonal projections of immunostained euploid (n=6) and aneuploid (n=11) embryos for DNA (white), DRAM1 (turquoise), and NANOG (green). T=Trisomy. Source: Experiment 3. (**t**) DRAM1 mean cytoplasmatic intensity in TE and ICM of euploid and aneuploid cells. Euploid TE (n=312 cells from 6 embryos). Aneuploid TE (n=434 cells from 11 embryos). Euploid ICM (n=43 cells from five embryos). Aneuploid ICM (n=70 cells from 10 embryos). Each dot represents the mean of all cells per embryo. Unpaired t-test, *p=0.0267, ns = non-significant. Source: Experiment 3. Embryo sources are indicated in each section. Brightfield pictures were obtained during confocal imaging. All scale bars are 20 μm. Box and whisker plots show median, and whiskers show minimum to maximum values. Bar plots and scatter plots show mean ± s.d.

The online version of this article includes the following source data and figure supplement(s) for figure 3:

**Source data 1.** Raw data of figures.

**Figure supplement 1.** S15-p53 expression in trophectoderm (TE) vs OCT4-positive cells (inner cell mass, ICM/epiblast, EPI) in aneuploid embryos.

**Figure supplement 2—source data 1.** Raw data of figures.

One of the key activated pathways identified during the transcriptome analysis was p53-signaling. Thus, we sought to validate p53 activation by staining for the phosphorylation of Serine-15 on p53, which is phosphorylated upon stimuli such as DNA-damage and metabolic stress (*Loughery et al., 2014*). We found that indeed aneuploid embryos showed a higher percentage of cells with activated p53 (*Figure 3M–N*). These differences did not reach significance when considering the ICM/EPI-OCT4-positive cells or TE OCT4-negative cells separately (*Figure 3—figure supplement 1A–C*) probably due to insufficient statistical power.

We then stained an additional batch of embryos (*Supplementary file 6*) for the double-strand break marker γH2AX, with the aim of determining if the p53 activation was DNA-damage mediated. As a control, we treated human embryos with the DNA-damage inducer Bleomycin and compared them to untreated embryos. All cells of Bleomycin-treated embryos showed a pan-nuclear γH2AX pattern, while untreated embryos rarely showed cells containing foci (*Figure 3O*). Euploid and aneuploid embryos showed similar fractions of cells with γH2AX foci or pan-nuclear staining (*Figure 3P–Q*), which did not change when dividing the aneuploid embryos into high- and low-dosage groups, (*Figure 3R*).

Lastly, the transcriptome analysis showed significant upregulation in aneuploid and high-dosage aneuploid embryos of the gene *DRAM1* (DNA Damage Regulated Autophagy Modulator-1, *Figure 2B*). DRAM1 is in the lysosomal membrane, and it increases autophagic flux and apoptosis after stress-induced p53 activation in cancer cells (*Crighton et al., 2006*; *Guan et al., 2015*). We stained euploid and aneuploid embryos (*Supplementary file 6*) for DRAM1 and NANOG and found that the expression of DRAM1 is increased specifically in the cytoplasm of aneuploid TE cells but not in the ICM (NANOG-positive cells, *Figure 3S, T*).

## Aneuploidy increases apoptosis in the trophectoderm and impairs lineage segregation events

We next investigated whether the stress responses to aneuploidy are cell-type specific, as shown in the mouse (*Bolton et al., 2016*; *Singla et al., 2020*) and human 2D gastruloids (*Yang et al., 2021*). Concretely, we aimed to study the consequences of aneuploidy during the first lineage segregation event to TE and ICM, and the second lineage segregation event to EPI and PrE.

We first immunostained euploid and aneuploid 5-6dpf human embryos for the pluripotency marker NANOG (*Supplementary file 6*). NANOG protein was enriched in the nuclei of aneuploid TE and ICM cells (*Figure 4A–B*), which could suggest a delay in the exit from pluripotency in both lineages. We then co-stained 6-7dpf euploid and aneuploid blastocysts for OCT4, CASP3/7, and LC3B (*Supplementary file 7*). Overall, we found lower cell counts in both the TE (OCT-negative cells) and in the OCT4-positive cells (ICM/EPI cells, *Figure 4C–E*) of aneuploid embryos. This resulted in lower total cell numbers in aneuploid embryos (*Figure 4F*), in line with our previous observation (*Figure 3E*). We found a higher percentage of nuclear CASP3/7 positive cells (*Figure 4G–H*) as well as a higher CASP3/7 mean intensity per cell (*Figure 4G1*) specifically in the TE of aneuploid embryos. In contrast, not a single OCT4-positive cell of either the euploid or aneuploid embryos showed a nuclear CASP3/7 signal (*Figure 4G*). Aneuploid embryos also showed a higher number of CASP3/7-positive micronuclei in the cells (*Figure 4J–K*). The lineage-specific response to aneuploidy also extended to the presence of autophagy. The staining for LC3B showed that aneuploid embryos displayed overall increased levels of autophagy which was mainly taking place in the TE and not the ICM (*Figure 4L–O*).

Finally, we observed that the counts of OCT4-positive cells in the aneuploid embryos were distributed in two groups: one with very low counts and one with counts in the range of that of euploid embryos (*Figure 4E*). This led us to hypothesize that some of the aneuploid embryos had a delayed ICM development, with impairment of the second lineage segregation event. All blastocysts analyzed above were re-stained for GATA4 (PrE marker) and imaged together with OCT4 (ICM/EPI marker). We found that while all euploid embryos contained GATA4-positive cells, 39% of aneuploid embryos had no GATA4-positive cells (*Figure 4P–Q*). Furthermore, in those aneuploid embryos that contained GATA4-positive PrE cells, there was a significantly lower ratio of GATA4-positive to OCT4-positive/ GATA4-negative cells (*Figure 4R*). Categorization of the aneuploid embryos into a low- and a high-dosage group (*Supplementary file 8*) showed that the higher the number of imbalanced loci, the lower the cell numbers in the TE and PrE but not in the EPI (*Figure 4S*).

Taken together, these results suggest that there are important differences in how the embryonic lineages respond to aneuploidy. While the TE is activating autophagy and apoptosis, this is not the case for the ICM/EPI. In turn, while in both lineages, aneuploidy appears to promote a delay in the downregulation of NANOG during differentiation, the ICM of high-dosage aneuploid embryos frequently has poor or no differentiation to PrE.

## Discussion

In this work, we studied the consequences of uniform complex aneuploidy in human preimplantation embryos. In our study, we found that, while aneuploid embryos had lower cell numbers in both TE and ICM/EPI, the effect of aneuploidy was cell-type dependent. Aneuploid TE presented transcriptomic signatures of p53 signaling and apoptosis, and immunostaining of whole embryos showed that aneuploidy results in increased proteotoxic stress, autophagy, activated p53 independent from DNA-damage, and subsequent apoptosis. In contrast, in the ICM, aneuploidy affects PrE formation.

Overall, our gene-expression results are in line with previously published RNA-sequencing studies on aneuploid human embryonic cells that identified transcriptional signatures consistent with an alteration of cell proliferation (*Martin et al., 2023*; *Maxwell et al., 2022*; *Starostik et al., 2020*), the cell cycle (*Fuchs Weizman et al., 2019*; *Licciardi et al., 2018*; *Martin et al., 2023*; *Starostik et al., 2020*; *Vera-Rodriguez et al., 2015*), deregulation of autophagy (*Licciardi et al., 2018*; *Sanchez-Ribas et al., 2019*), p53 signaling (*Licciardi et al., 2018*) and apoptosis (*Groff et al., 2019*; *Licciardi et al., 2018*; *Martin et al., 2023*; *Maxwell et al., 2022*; *Sanchez-Ribas et al., 2019*).

We found that the strength of these transcriptomic alterations is dependent on the number of imbalanced genes. These gene-dosage-dependent effects of aneuploidy are akin to those extensively reported in human and mouse aneuploid cancer cell lines (*Dürrbaum and Storchová, 2016*). We

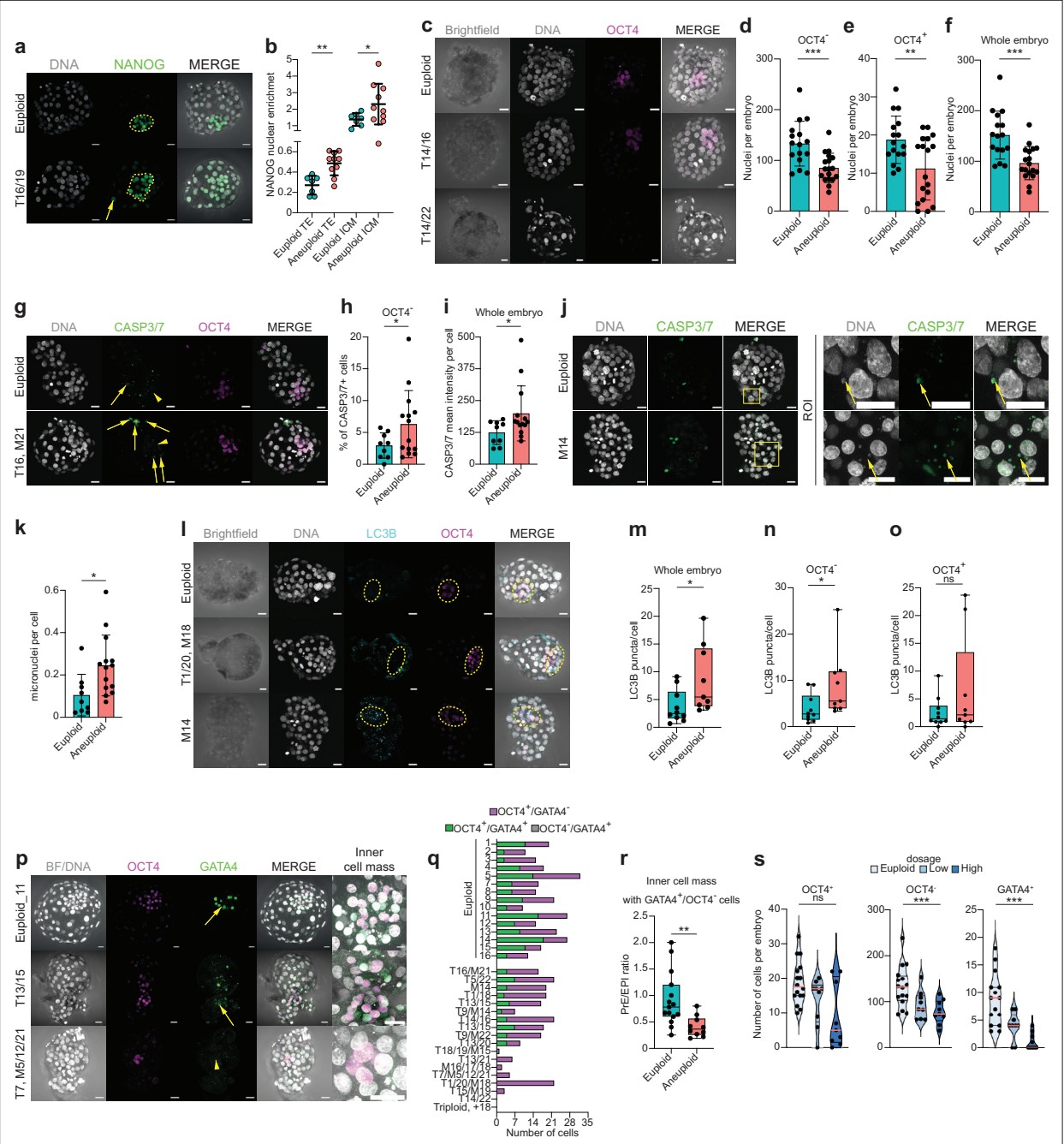

**Figure 4.** Aneuploid human embryos show less cells in trophectoderm and OCT4-positive cells and impaired lineage segregation events. (**a**) Orthogonal projections after immunostaining of euploid (n=7) and aneuploid embryos (n=10) for DNA (white) and NANOG (green). T=Trisomy. Source: Experiment 3. (**b**) NANOG nuclear enrichment in cells of the trophectoderm (TE) and inner cell mass (ICM) of euploid and aneuploid embryos. Each dot represents the mean of all cells per embryo. Euploid TE (n=718 cells from seven embryos). Aneuploid TE (n=708 cells from 10 embryos). Euploid ICM (n=77 cells from seven embryos). Aneuploid ICM (n=77 cells from 10 embryos). For euploid TE versus aneuploid TE: Unpaired t-test, **p=0.0011. For euploid versus aneuploid ICM: Unpaired t-test with Welch's correction, *p=0.0474. (**c**) Orthogonal projections after immunostaining of euploid and aneuploid embryos for DNA (white) and OCT4 (magenta). The first aneuploid panel (Trisomy 14 and 16) shows a similar number of ICM cells compared to the euploid embryo. The second aneuploid panel (Trisomy 14 and 22) shows an embryo without an ICM. T=Trisomy. Source: Experiment 4. (**d,e,f**) Differences in number of nuclei per embryo between euploid and aneuploid embryos in the (**d**) OCT4-negative cells (trophectoderm) ***p=0.0006, (**e**) OCT4-positive cells (ICM/epiblast, EPI), **p=0.0055 and (**f**) whole embryo ***p=0.0003. Euploid n=16, Aneuploid n=18. Unpaired t-test. Source: Experiment 4. (**g**) Orthogonal projections after immunostaining of euploid and aneuploid embryos for DNA (white), CASP3/7 (green), and OCT4 (magenta). c. T=Trisomy, M=Monosomy. Yellow arrows indicate presence of the signal and yellow arrow heads indicate absence of signal. (**h**) Percentage (%) of CASP3/7 positive (+) cells in the trophectoderm lineage (OCT4-negative cells). Euploid n=9, Aneuploid n=14. Unpaired t-test with Welch's

*Figure 4 continued on next page*

*Figure 4 continued*

correction, *p=0.0453. Source: Experiment 4. (**i**) CASP3/7 mean intensity per cell of whole embryos. Euploid n=9, Aneuploid n=14. Unpaired t-test with Welch's correction, *p=0.0332. Source: Experiment 4. (**j**) Left panel: Orthogonal projections after immunostaining of euploid and aneuploid embryos for DNA (white) and CASP3/7 (green) showing micronuclei. Yellow box indicates zoomed-in region of interest (ROI). Yellow arrows indicate presence of micronuclei and overlap between DNA and CASP3/7. Source: Experiment 4. (**k**) Number of micronuclei per cell in euploid and aneuploid whole embryos. Euploid n=9, Aneuploid n=14. Unpaired t-test with Welch's correction, *p=0.0115. Source: Experiment 4. (**l**) Orthogonal projections after immunostaining of euploid and aneuploid embryos for DNA (white), LC3B (turquoise), and OCT4 (magenta). Yellow circle indicates the ICM. The first aneuploid panel (Trisomy 1 and 20 and monosomy 18) shows OCT4-positive cells (ICM/EPI) with low levels of autophagy. The second aneuploid panel (Monosomy 14) shows an embryo with high levels of autophagy in the ICM. Source: Experiment 4. (**m, n, o**) Differences in LC3B puncta per cell between euploid (n=10) and aneuploid (n=9) embryos in the (**m**) whole embryo, *p=0.0220, (**n**) OCT4-negative cells (trophectoderm),*P=0.0172 and (**o**) OCT4-positive cells (ICM/EPI) ns = non-significant. Mann-Whitney test. Source: Experiment 4. (**p**) Orthogonal projections after immunostaining of euploid and aneuploid embryos for DNA (white), OCT4 (magenta), and GATA4 (green). The first aneuploid panel (trisomy 13 and 15) shows an embryo with presence of GATA4 positive cell (PrE). The second aneuploid panel (Trisomy 7 and monosomies 5, 12, and 21) shows an embryo that did not contain GATA4 positive cells. Source: Experiment 4. (**q**) Differences in the number of cells per embryo that were OCT4 positive and either GATA4 negative (magenta) or positive (green) or OCT4 negative and GATA4 positive (gray) between euploid (n=15) and aneuploid (n=18) embryos. In case GATA4-positive cells (PrE) were present we considered the GATA4-negative cells to be epiblast (EPI). All euploid embryos contained GATA4-positive cells, 7/18 aneuploid embryos had an ICM completely lacking GATA4-positive cells (Fisher-exact test, p=0.009). (**r**) Differences in the PrE/EPI ratio between euploid and aneuploid embryos that had an ICM containing GATA4-positive cells. Euploid n=15, Aneuploid n=10. Unpaired t-test, **p=0.0068. Source: Experiment 4. (**s**) Violin plots with box plots of OCT4-positive-cells (ICM/EPI) per embryo (left, ns), OCT4-negative cells (trophectoderm) per embryo (center, ***p<0.001), and GATA4-positive-cells (PrE) per embryo (right, ***p<0.001, Jonkheere-Terpstra test) depending on gene dosage. Euploid (light blue), Low-dosage (middle blue), and High-dosage (darkest blue). Source: Experiment 4. Brightfield pictures were obtained during confocal imaging. All scale bars are 20 μm. Box and whisker plots show median and minimum to maximum values. Bar plots show mean ± s.d. Besides (**b**) all dots represent one embryo.

The online version of this article includes the following source data for figure 4:

**Source data 1.** Raw data of figures.

found that higher numbers of aneuploid loci led to stronger induction of p53 target genes and apoptosis responses in the TE cells. Interestingly, high-dosage aneuploid embryos displayed an inhibited unfolded protein response, as also seen in aneuploid embryonic cells by another study (*Gallardo et al., 2023*), while this pathway was activated in low-dosage aneuploid embryos. These results suggest that while low-dosage aneuploid cells are trying to restore homeostasis by the unfolded protein response (*Hetz et al., 2020*; *Jäger et al., 2012*; *Li and Zhu, 2022*), high-dosage aneuploid cells are past this point and undergo apoptosis by p53 activation.

In the subsequent experiments, we found that aneuploid embryos showed increased signals of early (CASP8) and late (CASP3/7) caspases, indicating activation of the extrinsic apoptotic pathway (*McIlwain et al., 2013*). The higher levels of p62 and LC3B we found in these embryos suggests that this activation of apoptosis results at least in part from a sustained interaction between p62, LC3B, and CASP8 (*Pan et al., 2011*). In parallel, aneuploid embryos showed evidence of an activated p53 pathway, not only on a transcriptomic level, but also by presenting elevated levels of Serine 15 phosphorylated p53 (*Loughery et al., 2014*). This p53 activation has been previously observed on a transcriptomic level in another study on human embryos (*Licciardi et al., 2018*) and was identified as a key mediator of aneuploidy-dependent apoptosis in the mouse (*Singla et al., 2020*). While ATM-mediated phosphorylation of p53 on S15 has been reported in cancer cell lines as a consequence of DNA damage during chromosome missegregation (*Janssen et al., 2011*), we found no evidence of increased levels of DNA-damage in our cohort of embryos. It is worth noting that studies on constitutionally aneuploid cell lines have shown contradicting results, with some studies finding p53 activation (*Li et al., 2010*; *Thompson and Compton, 2010*) and others not (*Tang et al., 2011*). An alternative route for p53 activation is through p38, a sensor for cellular stress such as endoplasmic reticulum stress (*Mishra and Karande, 2014*; *Thompson and Compton, 2010*). Metabolic alterations might also promote ATM-mediated activation of p53 (*Li et al., 2010*) or phosphorylation of Serine 15 (*Jones et al., 2005*). Our study also identified *DRAM1* as a significantly upregulated gene in aneuploid embryos. *DRAM1* is directly regulated by p53 and is not only able to regulate autophagic flux but it also regulates apoptosis, making it likely an important downstream effector of the p53-mediated effects of aneuploidy in human embryos (*Crighton et al., 2006*; *Guan et al., 2015*). Lastly, it is worth considering that the complex abnormal human embryonic cells present in our sample might have been subjected to mitotic stress, as suggested by the enrichment of mitotic spindle pathway genes observed in the RNA-seq dataset. Mitotic stress might be induced by sustained proteotoxic stress

(*Zhu et al., 2018*) and can further have contributed to p53 activation. Based on this evidence, we propose that the p53 activation we observed in the embryos with presumed meiotic aneuploidy is akin to that of constitutionally aneuploid cells, and not due to DNA-damage but to cellular stress caused by proteotoxic stress. Taken together, the increased levels of apoptosis we find in aneuploid embryos appear to be mediated at least by two pathways: the sustained p62/LC3B mediated response to unfolded proteins and by p53 activation as a response to cellular stress. An interesting observation from the apoptosis stainings is that aneuploid embryos contained increased numbers of CASP3/7 positive micronuclei. This is similar to findings in a non-human primate model showing that abnormal chromosomes can be encapsulated in micronuclei, which can subsequently be selectively eliminated from the embryo (*Daughtry et al., 2019*). The presence of DNA in the cytoplasm has also recently been observed during live-imaging of human embryos, which occurs during blastocyst expansion due to nuclear DNA shedding (*Domingo-Muelas et al., 2023*). In human cleavage stage embryos these micronuclei have been shown to contain genetic material that originates from chromosome breakages due to replication fork stalling and DNA-damage (*Palmerola et al., 2022*).

In the second part of our work, we focused on studying the lineage-specific response to complex aneuploidy, as the studies on reversine-treated mouse embryos (*Bolton et al., 2016*; *Singla et al., 2020*) and human 2D gastruloids (*Yang et al., 2021*) also identified a cell-type specific effect of aneuploidy. In the mouse, aneuploid cells are eliminated by apoptosis in the EPI, while tolerated in the TE, and in human 2D gastruloids, aneuploid cells are eliminated in the post-gastrulation embryonic germ layers but tolerated in the TE. In the human embryo, the first lineage segregation is established at 5dpf, and the ICM and TE are marked by the differential expression of NANOG and GATA3 (*Gerri et al., 2020*; *Meistermann et al., 2021*; *Allègre et al., 2022*; *Regin et al., 2023*). We found that both aneuploid ICM and TE cells showed nuclear enrichment of NANOG, which can be suggestive of a delay or impairment of the first lineage segregation event, possibly due to failure or delay in the exit from pluripotency (*Zhang et al., 2016*).

While we found increased apoptosis in the aneuploid TE, we found that the ICM/EPI had no CASP3/7 positive cells, despite showing lower cell numbers than the euploid ICM/EPI. These results are in line with the findings of Martin and colleagues, where the RNA-sequencing of TE and ICM samples showed disrupted regulation of apoptosis in mosaic aneuploid TE cells while the ICM showed primarily deregulated mitochondrial function (*Martin et al., 2023*). In contrast, while the study of Victor et al also finds increased apoptosis in the aneuploid TE, some of their embryos did present CASP3/7 positive signals in the ICM. Further research will be needed to understand if the differences with our findings and those of Martin et al are related to technical or to biological factors. A potential explanation for the lack of apoptosis in the human ICM/EPI could be due to an uncoupling between aneuploidy-induced stress signals and apoptosis, as observed in human embryonic stem cells (*Mantel et al., 2007*). In this context, the lower cell numbers in the aneuploid ICM/EPI could be explained by other known effects of *p53* activation next to the induction of apoptosis. First, *p53* activation can decrease cell proliferation, leading to lower cell numbers. Second, *p53* activation can downregulate *NANOG* expression (*Abdelalim and Tooyama, 2014*; *Lin et al., 2005*), potentially leading to loss of pluripotency and driving the cells to differentiation and out of the ICM/EPI.

Another observation was that while the aneuploid TE had increased levels of autophagy markers and DRAM1 expression, this did not reach statistical significance in the ICM/EPI. This aligns with work on human embryos showing the absence of activation of p53-signaling and autophagy in the transcriptome of the ICM of mosaic human embryos (*Martin et al., 2023*), but is in contrast with the reversine-treated mouse embryos and human gastruloids, showing increased autophagy and p53 signaling in the ICM lineages (*Singla et al., 2020*; *Yang et al., 2021*). Next to a species-specific effect, it is possible that the difference between the mouse, human gastruloid and human studies are because the reversine treatment used had direct effects on p53 (*D'Alise et al., 2008*; *Santaguida et al., 2010*), leading to the activation of autophagy across the entire cells.

Finally, we found that in those embryos that had already undergone the second lineage segregation event, the aneuploid embryos had a significantly lower ratio of PrE (GATA4-positive) to EPI (OCT4-positive/GATA4-negative) cells than euploid embryos, suggesting that aneuploidy has a negative effect on PrE formation. Interestingly, recent work comparing the gene-expression of euploid and mosaic aneuploid embryos has also found that euploid embryos have a higher expression of the PrE marker *SOX17*, (*Martin et al., 2023*).

While in mice, apoptosis is common during PrE formation to eliminate misallocated cells through cell competition (*Hashimoto and Sasaki, 2019*; *Plusa et al., 2008*), our data and that of others suggest that the human ICM rarely undergoes apoptosis, even in karyotypically aberrant cells (*Martin et al., 2023*; *Victor et al., 2019*; *Yang et al., 2021*). While a possible explanation suggested by our NANOG stainings is that in aneuploid human embryos the PrE forms later due to retention of NANOG in the ICM and TE, the mechanisms behind this second lineage differentiation defect remain to be elucidated.

When interpreting the results of this study, it is important to bear in mind that we identified the ICM/EPI as staining positive for OCT4 and the PrE as positive for GATA4. We confirmed that the embryos analyzed were expanded blastocysts which had undergone the first lineage segregations, as OCT4 was restricted to the inner cells and absent in TE cells (*Niakan and Eggan, 2013*). The identity of human ICM and EPI cells is currently under vigorous debate (*Meistermann et al., 2021*; *Radley et al., 2023*) and exclusive markers of either cell type are untested. Therefore, we cannot assign ICM or EPI identity to OCT4-positive cells with certainty. However, in blastocysts with GATA4-positive cells, the second lineage segregation had occurred and PrE was established, and consequently, we confidently assigned the EPI identity to OCT4-positive cells in these blastocysts. In GATA4-negative embryos OCT4-positive cells could be either ICM or EPI cells. Interestingly, we only found a single GATA4-positive cell lacking OCT4, supporting the hypothesis that PrE cells differentiate from EPI cells (*Meistermann et al., 2021*).

Our study has several limitations and there are some considerations to bear in mind. In this work, we did not analyze the transcriptome of the ICM, because the remaining embryos were used for immunostaining after TE biopsy, and we can, therefore, not draw any conclusions on the ICM transcriptome. We did not study embryos with the same type of aneuploidy and can thus not provide insight on chromosome-specific effects. Although some of our embryos carried genetic abnormalities in mosaic form as part of their complex karyotype (aneuploid/aneuploid mosaics), we did not study diploid/aneuploid mosaic embryos and can, therefore, not assert that these also activate the same cellular stress pathways in their aneuploid cells. Since these pathways are common to most cell types, our prediction would, however, be that they are activated in mosaic embryos too. The possibilities for functional studies and lineage tracing experiments in human embryos are very limited, which is why we can only present an observational study. Alternatively, in silico modeling data could be leveraged to address the roles of aneuploidy in blastocyst formation and development (*Nissen et al., 2017*). The gene-expression analysis revealed that embryos with naturally acquired and induced aneuploidies had few deregulated pathways in common, with mainly the mitotic spindle genes being differentially expressed in both. Conversely, we used reversine to induce aneuploidy in the immunostaining experiments to increase the sample size, and the results showed that the reversine-induced aneuploidies elicited the same stress responses as aneuploidy. This suggests that while reversine-based models are very close, they may not fully mimic endogenous aneuploidy. An alternative is to use the SAC-inhibitor A3146, recently proposed by the Zernicka-Goetz group to induce low-degree aneuploidy in mouse embryos (*Sanchez-Vasquez et al., 2023*). Finally, our results on the activation of autophagy and proteotoxic stress are also limited by experimental constraints. We cannot rule out indirect activation of apoptosis mediated by autophagy due to the degradation of cell organelles (*Gump and Thorburn, 2011*). Also, other upstream regulators of autophagy could be involved, such as the cGAS-STING pathway (*Krivega et al., 2021*). While we and others *Singla et al., 2020* used HSP70 as a marker for chronic protein misfolding, we cannot preclude that is also acts as a (temporary) suppressor of the extrinsic apoptosis pathway in aneuploid cells (*Gabai et al., 2002*; *Lanneau et al., 2008*).

Taken together, our results show that while the same pathways appear to be activated in human and mouse embryos with complex aneuploidy, human embryos have a dosage-dependent and differential response when it comes to the downstream effect of these pathways in the different cell lineages. While the mouse readily depletes aneuploid cells by apoptosis specifically from the embryonic lineage before implantation (*Bolton et al., 2016*; *Singla et al., 2020*), in the human embryo, apoptosis appears to be more active in the TE and aneuploidy negatively affects PrE formation in the ICM. Our results provide insight into the mechanisms by which human embryos respond to gene-dosage imbalances and may contribute to understanding how aneuploid cells are selectively eliminated in mosaic preimplantation embryos. Further research, especially by using human embryos with endogenous chromosomal mosaicism, will be needed to shed light on the interactions between proteotoxic stress,

the p53 pathway, and differentiation during early development, for which the most novel models of human embryo implantation (*Deglincerti et al., 2016*; *Kagawa et al., 2022*; *Shahbazi et al., 2020*; *Shahbazi et al., 2016*) will prove to be of incalculable value.

# Methods

**Key resources table**

| Reagent type (species) or resource | Designation | Source or reference | Identifiers | Additional information |
|---|---|---|---|---|
| Antibody | Anti-HSP70 (Mouse monoclonal) | Invitrogen | MA3-007 | IF(1:200) |
| Antibody | Anti-SQSTM1/p62 (Mouse monoclonal) | Abcam | ab56416 | IF(1:100) |
| Antibody | Recombinant Anti-LC3B (Rabbit monoclonal) | Abcam | ab192890 | IF(1:838) |
| Antibody | Anti-OCT3/4 (Mouse monoclonal) | Santa-Cruz | sc-5279 | IF(1:200) |
| Antibody | Anti-human phosphor-p53 S15 (Rabbit polyclonal) | R&D | AF1043 | IF(1:100) |
| Antibody | Anti-GATA4 (Rat monoclonal) | Invitrogen | 14-9980-82 | IF(1:400) |
| Antibody | Anti-DRAM (Rabbit polyclonal) | Invitrogen | PA20-335 | IF(1:500) |
| Antibody | Anti-phospho-histone H2A.X Ser139 (Mouse monoclonal) | Sigma-Aldrich | 05–636 | IF(1:500) |
| Antibody | Anti-NANOG (Goat polyclonal) | R&D | AF1197 | IF(1:200) |
| Chemical compound, drug | Reversine | Stem Cell Technologies | # 72614 | 0.5 µM |
| Other | CaspGLOW Fluorescein Active Caspase-8 Staining Kit | Invitrogen | 88-7005-42 | IF(1:300) |
| Other | CellEvent Caspase-3/7 Green Detection Reagent | Invitrogen | C10423 | IF(1:400) |

## Ethical approval

All experiments have been approved by the local Commission of Medical Ethics of the UZ Brussel (B.U.N. 143201628722) and the Federal Committee for Medical and Scientific Research on Human Embryos in vitro (AdV069 and AdV091). Patients from Brussels IVF (UZ Brussel) donated their embryos for research after written informed consent.

## Culture of human pre-implantation embryos

All human embryos, unless stated otherwise, were warmed using the Vitrification Thaw Kit (Vit Kit-Thaw; Irvine Scientific, USA) and cultured in 25 µL droplets of Origio Sequential Blast medium (Origio) at 37 °C with 5% $O_2$, 6% $CO_2$, and 89% $N_2$. Embryos were graded before vitrification by experienced clinical embryologists according to *Gardner and Schoolcraft, 1999*.

### Experiment 1

We warmed vitrified human blastocysts (5-6dpf) after PGT (28 euploid and 22 aneuploid) or at 3dpf (n=11). Post-PGT-embryos were left to recover for 2 hr in a culture medium, after which they underwent a second TE biopsy. For the reversine experiment, we cultured the 3dpf embryos for 24 hr in a culture medium supplemented with 0.5 µM reversine (Stem Cell Technologies), and after wash-out we let them develop until 5dpf and subsequently took a biopsy. The concentration of 0.5 µM reversine was chosen based on previous experiments in mouse embryos (*Bolton et al., 2016*; *Singla et al., 2020*).

### Experiment 2

We warmed 12 vitrified and undiagnosed human blastocysts (5-6dpf) of which six were cultured in culture medium for 16 hr. The other 6 embryos were cultured in medium supplemented with 0.1 mg/mL Bleomycin (Sigma-Aldrich) for 16 hr.

### Experiment 3

We warmed vitrified human blastocysts (5-6dpf) after PGT (7 euploid and 11 aneuploid) that were left to recover for 2 hr in culture medium.

### Experiment 4

We warmed 5dpf-6dpf PGT embryos (10 euploid and 14 aneuploid) and cultured them for 16 hr to ensure sufficient time to progress to the second lineage differentiation. Prior to fixation, we live-stained all the embryos with either Caspase-3/7 or Caspase-8 (*Supplementary file 9*) for 30 min. Additionally, we warmed 5dpf-6dpf PGT embryos (6 euploid and 4 aneuploid) and cultured them for 16 hr without live-staining.

## Biopsy procedure

The zona pellucida was opened (15–25 μm) on 4dpf using a Saturn Laser (Cooper Surgical). For PGT cycles, expanded good quality embryos were biopsied on 5dpf; early blastocysts were evaluated again for expansion on 6dpf. Briefly, embryos were transferred to HEPES buffered medium and biopsied on a Nikon inversed microscope equipped with a Saturn Laser. The herniated TE cells were then aspirated and cut using manual laser shoots. In case of sticking cells this was supported by mechanical cutting. After successful tubing, the blastocysts were vitrified using CBS-VIT High-Security straws (CryoBioSystem, L'Aigle, France) with dimethylsulphoxide-ethylene glycol-sucrose as the cryoprotectants (Irvine Scientific Freeze Kit, Newtownmountkennedy, County Wicklow, Ireland) (*Van Landuyt et al., 2011*). The TE biopsy for RNA-sequencing (Experiment 1) was performed with the same procedure, cells were biopsied from the same opening in the zona pellucida.

## PGT

During PGT, our Center of Medical Genetics at the UZ Brussels analyzed a TE sample with multiple cells for each embryo. Next Generation Sequencing: The chromosomal analysis was performed by WGA (Sureplex, Illumina) followed by library preparation (KAPA HyperPlus, Roche), sequencing on a NovaSeq (Illumina), and an in-house developed interpretation pipeline. The analysis has an effective resolution of 5 Mb.

### SNP-array

We relied on whole genome amplification (MDA, Repli-G) followed by genome-wide SNP array using Karyomapping (Vitrolife) with Bluefuse software (Vitrolife). In addition, SNP array data were analyzed with an in-house developed interpretation pipeline for aneuploidy detection.

## RNA-sequencing

Multiple (5-10) TE cells were used as input to generate full-length cDNA with the SMART-Seq™ v4 Ultra™ Low Input RNA Kit (Clontech Laboratories, Inc) according to the manufacturer's instructions. The quality of cDNA was checked using the AATI Fragment Analyzer (Advances Analytical). Library preparation was performed using the KAPA HyperPlus Library Preparation Kit (KAPA Biosystems) according to the manufacturer's instructions. During cDNA synthesis and library preparation we used 17 and 11 PCR cycles, respectively. Sequencing was performed on a NovaSeq 6000 (Illumina) with 25 million reads per sample.

## Bioinformatics analysis

Reads were trimmed using cutadapt version 1.11 to remove the 'QuantSEQ FWD' adaptor sequence. We checked the quality of the reads using the FastQC algorithm (*Love et al., 2014*). Count tables were generated using STAR (*Dobin et al., 2013*) (version 2.5.3) through mapping against the Genome Reference Consortium Human Build 38 patch release 10 (GRCh38.p10) combined with a general transfer format (GTF) file, both downloaded from the Ensembl database. The RNA-sequencing by Expectation Maximization (RSEM) (*Li and Dewey, 2011*) software (version 1.3.0) was used to produce the count tables.

Differential gene expression analysis was performed using R-studio (Version 1.1.456) with the edgeR (*Robinson et al., 2010*) and limma (*Ritchie et al., 2015*) packages. We included genes with

a count per million greater than 1 in at least two samples. The trimmed mean of M values (*Robinson and Oshlack, 2010*) algorithm was used for normalization. The normalized counts were then transformed in a $\log_2$ fold-change ($\log_2$FC) table with their associated statistics. In each comparison, genes with a | $\log_2$FC |>1 and an FDR <0.05 were considered as significantly differentially expressed.

Gene set enrichment analysis (GSEA) was performed using the GSEA software (version 4.3.2) from the Molecular Signature Database (https://www.gsea-msigdb.org/gsea/msigdb/index.jsp). We generated a ranked gene list based on the normalized count matrix of the whole transcriptome that was detected after differential gene expression. The ranked gene list was then subjected to the GSEA function, and we searched the Hallmark and C2 library for significantly enriched pathways. Significance was determined using a cut-off value of 25% FDR. This cut-off is proposed in the user guide of the GSEA (https://www.gsea-msigdb.org/gsea/doc/GSEAUserGuideTEXT.htm) especially for incoherent gene expression datasets, as suggested by our PCAs, which allows for hypothesis driven validation of the dataset.

The copy number variation (CNV) analysis was inferred from our RNA-sequencing data set using inferCNV R package (version 1.7.1) (Inferring copy number alterations from tumor single cell RNA-seq data https://github.com/broadinstitute/inferCNV/wiki). Our count table was used as input and compared to the reference set which in our case was the expression data of the euploid embryos. The analysis followed the standard workflow in inferCNV with the following parameters: 'denoise' mode, value of 1 was used for the 'cutoff,' prediction of CNV states using default Hidden Markov Models (HMM) method, 'cluster_by_groups' was set to TRUE, and other values were set by default.

## Gene dosage analysis

We calculated the total number of loci with imbalanced gene expression based on the genetic abnormalities identified during the PGT. For this, we used the number of coding loci per chromosomal region as listed in the ENSMBL database (ensemble.org) and included both gains and losses, either mosaic or homogeneously present, and both full chromosomal aneuploidy as well as segmental abnormalities. The embryos were ranked based on the total number of imbalanced loci and separated into a low- and a high-imbalanced gene dosage group. The low-dosage group contained the embryos with the lowest 50th percentile in number of unbalanced loci, the high-dosage group the embryos with 50th percentile or higher.

## Immunocytochemistry

Prior to fixation we removed the zona pellucida using pre-warmed to 37 °C Acidic Tyrode's Solution (Sigma-Aldrich) for 30 s–2 min and subsequently washed in PBS/2%BSA. Embryos were then fixed using 4% paraformaldehyde (Sigma-Aldrich) for 10 min, washed three times in PBS/2% BSA for 5 min, and then permeabilized using 0.1% Triton X-100 (Sigma-Aldrich) for 20 min followed by another washing cycle. We used PBS/2% BSA supplemented with 10% FBS for 1 hr to block non-specific protein binding. The embryos were then incubated overnight with primary antibodies (*Supplementary file 9*) diluted in blocking buffer at 4 °C. The next day, they underwent another washing cycle using PBS/2% BSA and were then incubated with secondary antibodies (1:200, *Supplementary file 9*) for 1 h at room temperature. After washing, nuclei were stained using Hoechst 33342 (5µg/mL, Life Technologies) diluted in PBS/2% BSA for 15 min. For imaging, we mounted the embryos on poly-L-lysine coated glass slides (Sigma-Aldrich) in 3 µL of PBS. To avoid flattening of the embryos we used Secure-Seal Spacers (9 mm diameter, 0.12 mm deep, Thermo Fisher) before putting the coverslips in place. For the experiments that required re-staining we imaged the embryos in 3 µL PBS/2%BSA in 18 well µ-Slides (Ibidi) and subsequently recuperated them. After recuperation, we photobleached the embryos stained for CASP3/7 by using maximum laser power of the 488 nm channel and maximum size of the pinhole for 10 min. The embryos were then re-stained using rat anti-GATA4 primary antibody and the matching Alexa Fluor 488 secondary antibody (*Supplementary file 9*) (to replace the CASP3/7 staining).

## Imaging and image analysis

Confocal imaging was performed using the LSM800 (Zeiss). Z-stacks were taken with a LD C-Apochromat 40 x/1.1 NA water immersion objective in optical sections of 1 µm. Nuclei were counted using

the Blob finder function of Arivis Vision 4D (3.4.0). All other measurements were performed using the Zen 2 (blue edition) or Image J software based on the optical sections and/or orthogonal projections.

## Statistics

The type of statistical test and p-values are mentioned in the figure legends. For most comparisons we used one-way ANOVA or Student's t-test with or without Welch's correction. For experiments that contained groups with small sample sizes, we used non-parametric tests (Mann-Whitney). Fisher-exact test was used to determine the dependency of the ploidy status on the presence or absence of primitive endoderm. We used GraphPad Prism 9.0.0 for statistical testing. The trend analysis regarding gene dosage effects was performed with SPSS using the Jonckheere-Terpstra test.

## Acknowledgements

The authors would like to thank Marleen Carlé from the Center for Medical Genetics (UZ Brussel) for assisting during the tubing of trophectoderm samples, Hanne Vlieghe (UCLouvain, Brussels) for the validation of antibodies and the reversine reagent, Wilfried Cools from the Biostatistics and Medical Informatics Group (VUB) for the statistical advice and the members of the BRIGHTcore facility (UZ Brussel) for performing the RNA sequencing. We thank Prof. Rajiv McCoy for the scientific input and data exchange. This study was funded by the Fonds for Scientific Research in Flanders -G017218N-(Fonds Wetenschappelijk Onderzoek – Vlaanderen [FWO]). M.R., E.C.D.D., and C.J. are doctoral fellows at the FWO with grant numbers (1133622 N, 1S73521N, and 11H9823N, respectively). Y.L. is a predoctoral fellow supported by the China Scholarship Council (CSC).

## Additional information

### Funding

| Funder | Grant reference number | Author |
| --- | --- | --- |
| Fonds Wetenschappelijk Onderzoek | 1133622N | Marius Regin |
| China Scholarship Council | | Yingnan Lei |
| Fonds Wetenschappelijk Onderzoek | 1S73521N | Edouard Couvreu De Deckersberg |
| Fonds Wetenschappelijk Onderzoek | 11H9823N | Charlotte Janssens |
| Fonds Wetenschappelijk Onderzoek | G017218N | Karen Sermon Claudia Spits |

The funders had no role in study design, data collection and interpretation, or the decision to submit the work for publication.

### Author contributions

Marius Regin, Conceptualization, Data curation, Formal analysis, Validation, Investigation, Visualization, Methodology, Writing – original draft, Project administration, Writing – review and editing; Yingnan Lei, Edouard Couvreu De Deckersberg, Software, Formal analysis, Visualization, Methodology, Writing – review and editing; Charlotte Janssens, Anfien Huyghebaert, Pieter Verdyck, Formal analysis, Methodology, Writing – review and editing; Yves Guns, Methodology, Writing – review and editing; Greta Verheyen, Resources, Methodology, Writing – review and editing; Hilde Van de Velde, Resources, Investigation, Methodology, Writing – original draft, Project administration, Writing – review and editing; Karen Sermon, Conceptualization, Resources, Supervision, Funding acquisition, Investigation, Methodology, Writing – original draft, Project administration, Writing – review and editing; Claudia Spits, Conceptualization, Resources, Data curation, Formal analysis, Supervision, Funding acquisition, Investigation, Visualization, Methodology, Writing – original draft, Project administration, Writing – review and editing

## Author ORCIDs

Marius Regin (iD) https://orcid.org/0000-0001-7053-7354
Claudia Spits (iD) https://orcid.org/0000-0002-0187-5138

## Ethics

All experiments have been approved by the local Commission of Medical Ethics of the UZ Brussel (B.U.N. 143201628722) and the Federal Committee for Medical and Scientific Research on Human Embryos in vitro (AdV069 and AdV091). Patients from Brussels IVF (UZ Brussel) donated their embryos for research after written informed consent.

Reviewer #2 (Public review): https://doi.org/10.7554/eLife.88916.3.sa1
Reviewer #3 (Public review): https://doi.org/10.7554/eLife.88916.3.sa2
Author response https://doi.org/10.7554/eLife.88916.3.sa3

# Additional files

## Supplementary files

- MDAR checklist
- Supplementary file 1. PGT diagnoses and inferCNV results first experiment.
- Supplementary file 2. Differentially expressed genes aneuploid versus euploid embryos.
- Supplementary file 3. Differentially expressed genes reversine versus euploid embryos.
- Supplementary file 4. Ranks of imbalanced gene loci for the first experiment.
- Supplementary file 5. Differentially expressed genes high-dosage versus euploid embryos.
- Supplementary file 6. Gene dosage ranking Experiment 3 and 4.
- Supplementary file 7. PGT diagnoses experiment 4.
- Supplementary file 8. Gene dosage classification experiment 4.
- Supplementary file 9. Antibodies and reagents used in this study.

## Data availability

The data supporting this manuscript are stored at the Open Science Framework (OSF) and can be accessed through the identifier: tc248. Raw sequencing data of human samples is considered personal data by the General Data Protection Regulation of the European Union (Regulation (EU) 2016/679), because SNPs can be extracted from the reads, and can therefore not be publicly shared. The count tables used in this study are shared on the OSF site, the raw sequencing data can be obtained from the corresponding author upon reasonable request and after signing a Data Use Agreement.

The following dataset was generated:

| Author(s) | Year | Dataset title | Dataset URL | Database and Identifier |
|---|---|---|---|---|
| Spits C, Verloes A, Regin M | 2024 | Complex aneuploidy triggers autophagy and p53-mediated apoptosis and impairs lineage segregation events in human preimplantation embryos | https://osf.io/tc248/ | Open Science Framework, tc248 |

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
