## [Editor Report · eLife Assessment]

This study provides **valuable** insights into the cellular responses to complex aneuploidy in human preimplantation embryos. The evidence supporting the claims of the authors is now **convincing** after addressing previous concerns. This work will be of interest to embryologists, geneticists and scholars working on reproductive medicine by increasing our understanding of how human embryos respond to chromosomal abnormalities.

---

## [Referee Report · Reviewer #2 (Public review)]

A high fraction of cells in early embryos carry aneuploid karyotypes, yet even chromosomally mosaic human blastocysts can implant and lead to healthy newborns with diploid karyotypes. Previous studies in other models have shown that genotoxic and proteotoxic stresses arising from aneuploidy lead to the activation of the p53 pathway and autophagy, which helps eliminate cells with aberrant karyotypes. These observations have been here evaluated and confirmed in human blastocysts. The study also demonstrates that the second lineage and formation of primitive endoderm are particularly impaired by aneuploidy.

Comments on revisions:

The authors have addressed the critical issues sufficiently. In particular, they improved the data analysis and added additional data from embryonal samples.

---

## [Referee Report · Reviewer #3 (Public review)]

This study provides valuable insights into the cellular responses to complex aneuploidy in human preimplantation embryos. The authors have significantly expanded their sample size and conducted additional analysis and experiments to address previous concerns. The revised manuscript presents stronger evidence for gene dosage-dependent effects of aneuploidy on stress responses and lineage segregation. Overall, the findings contribute important knowledge to our understanding of how human embryos respond to chromosomal abnormalities.

Overall, the revision has substantially improved the manuscript and addressed the major concerns raised in the initial review.

---

## [Author Response]

The following is the authors’ response to the original reviews.

**eLife Assessment**
This study has uncovered some important initial findings about cellular responses to aneuploidy through analysis of gene expression in a set of donated human embryos. While the study's findings are in general solid, some experiments lack statistical power due to small sample sizes. The authors should try to get much more insight with their data highlighting the novel findings.

We thank the editor for considering our manuscript for publication at *elife*, and for the helpful and thorough reviews of our work. Based on the suggestions of the reviewers, we have carried out additional experiments, expanded the sample size and reanalyzed the data. This has resulted in a thoroughly revised manuscript and much improved work, which we are convinced meets the requirements to be published as a version of record. Of note, the experiments for the revision required the support by 2 additional researchers from our lab which are now coauthors.

These are the main changes made to the initial manuscript:

(1) The RNA-seq data (Figures 1+2) is now FDR corrected and been reanalyzed. This has not affected the initial observations on the activation of p53 and apoptosis in aneuploid human embryos, as well as that the transcriptomic changes are driven by gene dosage effects.

(2) We have included the transcriptome analysis of reversine-treated embryos in the supplementary data.

(3) For validation of novel findings such as the presence of DNA-damage and the expression of DRAM1 in aneuploid embryos, we now include the stainings of 30 human blastocysts (Figure 3o-t). We found absence of DNA-damage in aneuploid embryos and that DRAM1 is increased in the TE but not the ICM of aneuploid embryos.

(4) We re-analyzed the co-expression of CASP8/HSP70 in reversine-embryos as suggested by reviewer 1 and found that both proteins tend to be co-expressed.

(5) We have added a new analysis of NANOG expression (Figure 4a,b) of the embryos used in Figure 3o-t and have found retention of NANOG protein in both the TE and ICM.

(6) We have added 6 euploid and 4 aneuploid embryos to Figure 4l-s, which support the conclusions on the absence of autophagy activation in the ICM and failure of PrE formation in aneuploid embryos.

(7) We have significantly changed the layout of the figures, revised the supplementary tables, added source data files and rewritten the discussion.

Regarding the sample size of the study, it is important to emphasize that human embryos are ethically sensitive material and that those with the specific genetic content we used in this study are rare, limiting our ability to expand the sample size. For the revision, we have added 40 human blastocysts to our initial 85 embryos. Compared to similar and high-quality studies using human embryos, our study shows a relatively large sample size (n=125): Victor *et al.* 2021: 30 human blastocysts for immunostainings1; Martin *et al.* 2023: 14 human blastocysts2; Martin *et al.* 2024: 64 human blastocysts3; Domingo-Muelas *et al.* 2023: 23 human blastocysts4.

**Public Reviews:**

**Reviewer#1(PublicReview):**
This study investigated an important question in human reproduction: why most fully aneuploid embryos is incompatible with normal fetal development. Specifically, the authors investigated the cellular responses to aneuploidy through analysis of gene expression in a set of donated human blastocysts. The samples included uniform aneuploid embryos of meiotic origin and mosaic aneuploid embryos from the SAC inhibitor reversine treatment. The authors relied mainly on low-input RNA sequencing and immunofluorescence staining. Pathway analysis with RNA-seq data of trophectoderm cells suggested activation of p53 and possibly apoptosis, and this cellular signature appeared to be stronger in TE cells with a higher degree of aneuploidy. Immunostaining also found some evidence of apoptosis, increased expression of HSP70 and autophagy in some aneuploid cells. With combinational OCT4 and GATA4 as lineage markers, it appeared that aneuploidy could alter the second lineage segregation and primitive endoderm formation in particular.Although this study is largely descriptive, it generated valuable RNA-seq data from a set of aneuploid TE cells with known karyotypes. Immunostaining results in general were consistent with findings in mouse embryos and human gastruloids.

We thank the reviewer for the thorough evaluation of our manuscript. We have implemented most of the suggestions, which have further strengthened the original findings.

While there is a scarcity of human embryo materials for research, the lack of single cell level data limits further extension of the presented data on the consequences of mosaic embryos.

We did not include single cell RNA-seq data of mosaic human embryos in our study because we focused on embryos diagnosed with complex meiotic abnormalities. Our hypothesis was that the cellular consequences of aneuploidy would be strongest in this type of aneuploidies and most evident to identify and would allow us to provide a basis for the mechanisms of elimination of aneuploid cells in human embryos. In the manuscript (lines 596-626) we acknowledge the limitations of the extrapolation of our results to mosaic embryos.

A major concern is that the gene list used for pathway analysis is not FDR controlled. It is also unclear how the many plots generated with the "supervised approach" were actually performed.

We agree with the concerns about the fact that our differential expression gene list was not FDR but p-value ranked. We followed the suggestion of the reviewer and revised the RNAseq analysis and focused primarily on pathway analysis. We have also added the comparison between aneuploid and reversine treated embryos to the supplementary data and expanded the analysis of high dosage and low dosage embryos. Importantly, the new analysis has not changed the original finding that aneuploid embryos show hallmarks of p53 activation and apoptosis, and that these effects are gene dosage dependent. The manuscript now includes two completely revised and new figures 1 and 2.

Since we discarded the data generated from our previous approach, we do not use the term supervised approach anymore.

The authors also appear to have ignored the possibility that high-dosage group could have a higher mitotic defect.

This is indeed a possibility. In the discussion (lines 504-508) we have now incorporated the notion that the high dosage embryos could have higher mitotic defects, although our data cannot provide any evidence for this. Of note, the gene expression data shows that all aneuploid embryos (including low dosage and reversine embryos) equally show an enrichment for mitotic spindle pathway genes.

Assuming a fully aneuploid embryo, why do only some cells display p53 and autophagy marker?

This is a very good question, on which we can only speculate, but the answer likely lies in the diversity across cells of the same embryo.

Even in genetically homogenous tissues and cell cultures, individual cells can exhibit different levels of stress responses, such as p53 activation and apoptosis. This variation may be influenced by the local cellular environment, stochastic gene expression, or differences in cell cycle stages. Other studies on fully aneuploid human embryos could also not detect apoptotic responses in every cell1,3.

For instance, p53 activation differs even between cells that have a similar number of DNA breaks, and this activation is influenced by both cell-intrinsic factors and previous exposure to DNA damage5.

Cell cycle tightly regulates the response of cells to different stressors. For instance, cells in G1 or S-phase might be more sensitive to apoptosis signals6, while those in G2/M might escape this response temporarily7. Autophagy is more induced in G1 and S phases, with reduced activity in G2 and M phases8.

Individual cells may also have different levels of success in the activation of the compensatory pathways, including the unfolded protein response, autophagy, or changes in metabolism, resulting in some cells adapting better than others.

The expression of p53 and the sensitivity to apoptosis could also be influenced by epigenetic differences between cells, which may alter their transcriptional response to aneuploidy. Even in a genetically identical population, cells can have different epigenetic landscapes, leading to heterogeneous gene expression patterns.

The conclusion about proteotoxic stress was largely based on staining of HSP70. It appears from Figure 3 d,h that the same cells exhibited increased HSP70 and CASP8 staining. Since HSP70 is known to have anti-apoptotic effect, could the increased expression of Hsp70 be an anti-apoptotic response?

Our conclusion about proteotoxic stress was not solely based on HSP70 expression. We also stained for LC3B and p62, which are markers for autophagy and when highly expressed indirectly point towards underlying proteotoxic stress in the cells.

We reanalyzed the imaging of the stainings in the reversine-treated embryos, and found that the same cells were positive for both HSP70 and CASP8 staining while the minority was single positive (shown now in Figure 3k,l).

HSP70 does indeed not only unfold misfolded and aggregated proteins but does also have a function during cell survival and apoptosis9. HSP70 has been for instance found to inhibit the cleavage of Bid through active CASP8 within the extrinsic apoptosis pathway10. It is thus possible that it temporarily plays this role, and we have acknowledged this in the discussion (lines 623-626). On the other hand, the evidence points at an active apoptosis in the TE, with concomitant cell loss, so if HSP70 is indeed having an anti-apoptotic effect, it is having a limited impact.

**Reviewer #2 (Public Review):**
A high fraction of cells in early embryos carry aneuploid karyotypes, yet even chromosomally mosaic human blastocysts can implant and lead to healthy newborns with diploid karyotypes. Previous studies in other models have shown that genotoxic and proteotoxic stresses arising from aneuploidy lead to the activation of the p53 pathway and autophagy, which helps eliminate cells with aberrant karyotypes. These observations have been here evaluated and confirmed in human blastocysts. The study also demonstrates that the second lineage and formation of primitive endoderm are particularly impaired by aneuploidy.This is a timely and potentially important study. Aneuploidy is common in early embryos and has a negative impact on their development, but the reasons behind this are poorly understood. Furthermore, how mosaic aneuploid embryos with a fraction of euploidy greater than 50 % can undergo healthy development remains a mystery. Most of our current information comes from studies on murine embryos, making a substantial study on human embryos of great importance. However, there are only very few new findings or insights provided by this study. Some of the previous findings were reproduced, but it is difficult to say whether this is a real finding, or whether it is a consequence of a low sample number. The authors could get much more insight with their data.

We thank the reviewer for the thorough evaluation of our manuscript and the valuable suggestions made in the private recommendations. We have expanded the sample size and have carried out additional experiments that have significantly improved the manuscript.

**Recommendations for the authors:**

**Reviewer #1 (Recommendations For The Authors):**
(1) Instead of using cut off to generate a list, the authors could just rank the entire detected transcriptome for GSEA. This method fits better the authors' intentions of "primarily focused on pathway analysis." The cut-off value "-log10(p-value)<0.05" is not correct. As we can see from the PCA plot, one would not expect many cut off defined DEGs at all. The most obvious transcriptome change is dosage dependent, as the authors cleared showed with InferCNV.

We thank the reviewer for this suggestion and agree that this was an important concern of the study. We have entirely revised the RNA-seq analysis based on the proposed approach (Figure 1 and 2, Supplementary Figure 1). Also, we have included the analysis of aneuploid versus reversine treated embryos, which has allowed us to determine the differences between naturally occurring chromosomal abnormalities and those that are induced using reversine (Supplementary Figure 1).

We first performed differential gene expression analysis using DESEq2 with a cut-off value for significantly differentially expressed genes of | log2FC | > 1 and an FDR < 0.05. Based on the PCAs and the low number of differentially expressed genes for all comparisons, besides high dosage versus euploid embryos, we focussed primarily on pathway analysis.

For that, based on the reviewer’s suggestion, we generated a ranked gene list using the GSEA software (version 4.2.2, MSigDatabase) based on the normalized count matrix of the whole transcriptome that was detected after differential gene expression. The ranked gene list was then subjected to the *run GSEA* function, and we searched the Hallmark and C2 library for significantly enriched pathways. Thus, we could generate normalized enrichment scores, allowing us to predict whether a pathway is activated or suppressed. The details of the new analysis are described in the Material and Methods section (lines 220-232). Significance was determined using a cut-off value of 25% FDR. This cut-off is proposed in the user guide of the GSEA (https://www.gsea-msigdb.org/gsea/doc/GSEAUserGuideTEXT.htm) especially for incoherent gene expression datasets, as suggested by our PCAs, which allows for hypothesis driven validation of the dataset.

Indeed, we found that the most important transcriptome changes are aneuploidy dosage dependent. High dosage embryos show signatures of cellular unfitness, while low-dosage embryos still seem to activate survival pathways (lines 349-364).

This new analysis did not only increase robustness of our results but also introduced novel findings, which pave the road for future studies.

The validity of our findings is supported by recent work by the Zernicka-Goetz lab. We found that hypoxia is upregulated in low dosage human aneuploid TE cells. In line with our data, the Zernicka-Goetz lab found in a mouse model of low degree chromosomal abnormalities that hypoxia inducible factor 1A (HIF1A) promotes survival of extraembryonic aneuploid cells by reducing levels of DNA damage11.

(2) It would be very helpful if the authors could perform co-staining of multiple stress markers to better understand the origins of apoptosis and autophagy cells. In Fig 3d and 3h, it seems that the same reversine treated embryo was stained with CASP8, LC3B and HSP70. Is there any correlation between CASP8 and HSP70 at the single cell level? Is there any correlation between p53 and LC3B as the authors suggested, possibly through DRAM1?

We decided to use the complex aneuploid embryos that were left at our facility for the validation of novel findings such as upregulation of DRAM1 and presence and consequences of DNA damage in aneuploid embryos. As suggested by the editor and the other reviewer we also added embryos to existing datasets to increase the sample size where necessary. Therefore, we did not include other co-staining’s of multiple stress markers.

Following the reviewer’s suggestion, we reanalyzed the existing stainings and evaluated whether there is a correlation between CASP8 and HSP70 at the single cell level. The reversine-treated embryos were the only embryo group that was co-stained for both CASP8 and HSP70. We quantified the percentage of cells that were single or double positive for CASP8 and HSP70 and found a higher proportion of double positive cells than to single positives. Therefore, we concluded that there is indeed a correlation between both proteins at the single cell level in reversine-treated embryos and included this data in Figure 3k,l.

During the experiments for the revision, we found that the DRAM1 protein was upregulated in the cytoplasm of TE cells but not in the ICM of aneuploid embryos (Figure 3s,t), which validates the findings of the gene expression analysis. This data also supports our findings that autophagy is active in aneuploid TE cells while not significantly increased in aneuploid pluripotent ICM cells. Unfortunately, we could not stain LC3B and DRAM1 in the same embryo because the antibodies were raised in the same species.

(3) While " the possibilities for functional studies and lineage tracing experiments in human embryos are very limited," the authors can leverage in silico modelling (ie, PMID: 28700688) to address the roles of aneuploidy in blastocyst formation and development. Is there any selfregulating mechanism underlying the ratios of PrE and EPI? Is apoptosis of ICM cells a natural process during PrE formation (PMID: 18725515)?

It is a very interesting proposal to use *in silico* modelling to address the roles of aneuploidy during human blastocyst formation and lineage segregation. Although this type of analysis would yield very important insights, we are not able to address this point of the revision due to lack of expertise for this type of analysis in our group, requiring setting up a collaboration with experts in this field. In the discussion we proposed that future studies can leverage our data to be carried out in *silico* modelling and cited the proposed article (lines 608-610).

On the second part of the question, we would like to discuss the differences between mouse and human embryo studies. Parts of this were included in the discussion on the possible mechanisms of PrE elimination.

Is there a self-regulating mechanism for EPI/PrE formation?

To extrapolate the knowledge on mouse development to human it is important to bear in mind that (1) human embryos are outbred, as compared to inbred super-fertile laboratory mouse strains and (2) the embryos are donated to research by subfertile couples, which could compromise the EPI/PrE ratios. For instance, Chousal and colleagues found that poor quality blastocysts have a reduced number of PrE cells12. In human embryos the proportion EPI and PrE cells is indeed highly variable (20%-60%) and while the number of EPI cells does not increase between dpf6 and 7, the number of PrE cells does grow13. We found a similar variable number of EPI and PrE in our study on the lineage segregation mechanisms in good quality human embryos, with an absolute number of EPI of 12.1±6.5 cells and 8.4±3.44 PrE cells14.

By comparison, in late mouse blastocysts, the ratio EPI/PrE cells is consistent (2/3)15. Overall, self-regulating mechanisms in the human embryo are not yet studied in detail due to the lack of possible functional testing.

Is apoptosis a natural process during PrE formation?

Yes, in mice apoptosis is a natural process during PrE formation to eliminate misallocated cells of the inner cell mass through cell competition16,17. Yet, in the human embryo there is no evidence of such mechanisms. Although apoptosis is present even in human blastocysts of good quality18, the origin of such apoptotic cells is now still shown, although suboptimal culture conditions are known to increase cellular fragmentation19. Conversely, our data and that of others1,2 supports the notion that the pluripotent inner cell mass in human embryos is more resistant to apoptosis than the trophectoderm, even in karyotypically aberrant cells.

(4) The "count tables generated from the raw data files" could not be found in the source data files.

This slipped to our attention, we have added now the count tables to the source data files. Our apologies.

(5) Citations on aneuploidy literature were not done in a fully scholarly manner. It appears that authors selectively cite previous papers that are in support of their hypothesis but left out those with alternative conclusions.

We apologize if we missed any literature that contradicts our findings, it is not intentional. We would be grateful if the reviewer could provide such references.

In the manuscript we describe the alignment and differences of key findings with several studies (listed below) and the limitations of our study are extensively described in lines 596626.

Our findings align with other work on these aspects:

- RNA-sequencing data2,20–26

- Gene dosage effects drive the transcriptome of the aneuploid human embryo27,28

- Aneuploid cells are cleared by sustained proteotoxic stress followed by p53 activation, autophagy and eventually apoptosis29–37.

- p53 is active in constitutional aneuploid cells38

- The ICM is less sensitive to apoptosis1,2

Our findings differ with other work on these points:

- p53 activation is independent from DNA-damage39

- p53 is active in constitutional aneuploid cells40,41

- Apoptosis is only present in the aneuploid TE of aneuploid cells in the embryo29,30,42

**Reviewer #2 (Recommendations For The Authors):**
Comments:(1) The main problem is that there is no substantial novelty. The authors look at previously identified factors affected by chromosome gains and losses, but none of the new one from their analysis. Anything what could be potentially novel is not carefully analyzed (e.g. the difference between reversine-treated and aneuploid samples, or new potential candidates) or explained. This is really a pity.

In the revision, we have further elaborated on the DNA damage aspect by staining for DNA double-stranded breaks and have validated DRAM1 as an activated downstream effector of p53. We have also added the analyses of the gene-expression of the reversine-treated embryos.

(2) Some of the general statements on aneuploidy are confusing and often borderline generalized. E.g. introduction line 106: "If this (proteotoxic stress) remains unresolved by the activation of autophagy..." I am not aware of any publication suggesting that autophagy resolves proteotoxic stress in aneuploid cells. Citations that replication stress causes DNA damage in aneuploid cells are wrong. This link was first shown by Passerini et al. in 2016. etc.

We have clarified these statements in the introduction and added the proposed citations on replication stress that causes DNA damage in aneuploid cells (lines 95-108).

(3) In the figures the authors show a representative image of aneuploid and diploid embryos. Given the aneuploid embryos have widely different karyotypes, it would be important to clarify which of the embryos has been actually shown. Similarly, in the heat maps it is not clear which line is which embryo. This would be very useful.

We added the karyotypes of the aneuploid embryos to the images in figure 3 and 4. Since the heatmaps were removed from the figures we added the karyotypes to the PCAs in all figures.

(4) The authors constantly state that aneuploid embryo accumulate more DNA damage, which is supported by some of their observations, e.g. the DNA damage response is upregulated. It would be great if they would validated this statements with testing some markers for DNA damage.

We agree with the reviewer that this was an important point and addressing it has revealed that our initial assumption was incorrect and has provided new interesting findings. From the revised RNA-seq analysis, we found only one pathway (DNA damage response TP53) to be activated in all aneuploid embryos (Fig.1e). The ATM pathway was also activated specifically in high-dosage embryos. Following this, we set to test if DNA damage was indeed increased in aneuploid embryos by staining for DNA double strand breaks with gH2AX.

First, we investigated the gH2AX expression in 5dpf embryos in which we induced DNAdamage with Bleomycin. We compared 6 untreated versus 6 Bleomycin treated human embryos (Fig. 3m) and found that gH2AX foci were rarely present in the untreated embryos and that all cells of the treated embryos showed a pan-nuclear gH2AX staining.

Second, we compared the presence of gH2AX foci in the TE (NANOG negative cells), ICM (NANOG positive cells) and the whole embryo of 7 euploid versus 11 aneuploid embryos. Interestingly, we found no differences in the number of gH2AX foci or pan-nuclear gH2AX nuclei between euploid and aneuploid embryos (Fig 3o). When dividing our aneuploid embryos into high and low dosage embryos we could also not account for differences. Our data now suggests that complex aneuploid human embryonic cells of meiotic origin do not contain more DNA-double strand breaks, precluding DNA-damage as the source of p53 activation. Last, in our previous experiment we found that phosphorylated S15p53 is increased in aneuploid embryos, supporting an active p53 pathway as suggested by our transcriptomic data. Since we could not find DNA-damage in aneuploid human embryos we speculate that p53 is phosphorylated on Serine15 through metabolic stress as suggested by Jones and colleagues43. We also argue that proteotoxic stress might induce p53 expression as proposed by Singla and colleagues29.

(5) The source of embryos is only partially described in a figure legend. This should be expanded and described in the Materials and Methods section. The embryos are named, but this is nowhere explained. One can only assume that T is for trisomy and M is for monosomy.

We have divided the embryos into different experimental series (Experiment 1-4). This is now described in the Material and methods section (lines 157-175). Also, we have added the experiment number of each embryo to the supplementary tables and to the source data. The abbreviation for T = Trisomy and M = Monosomy was initially introduced in the last paragraph of the figure legend of figure 4. We now added it to every panel.

(6) Recent works from non-embryonic cells suggest that the cellular response to monosomy is different than the response to trisomy. Did the authors try to test this possible difference? For example, one could compare embryos M174/21, M2/19 and M17 with T2/10, T10/22 and T1/15/18/22.

We thank the reviewer for pointing this out. Our RNA-seq. dataset consisted of three embryos that contained trisomies only and four embryos that contained monosomies only. When reanalyzing our data we found different transcriptomic responses between monosomic only and trisomic only cells. Compared to euploid cells, monosomy only cells activate mainly the p53pathway and protein secretion while translation, DNA replication, cell cycle G1/S, DNA synthesis and processing of DNA double strand breaks were inhibited. Trisomy only cells show activated oxidative phosphorylation, ribosome and translation while protein secretion, apoptosis and cell cycle are inhibited. These differences were confirmed by testing transcriptomic differences between trisomic versus monosomic cells. Our results are similar to studies on human embryos20,26 and other monosomic and trisomic cell lines44,45. However, the interpretation of these results is very limited by the small sample size and the comparison of monosomies and trisomies of different chromosomes. Thus, we decided to keep this analysis out of the manuscript.

On the protein level, next to the small sample size, our results were also limited by the fact that not all embryos were stained with the same combinations of antibodies. LC3B was the only protein for which all embryos were immunostained. Thus, other protein data could not be re-analyzed due to even lower sample sizes.

Below we have separated the LC3B puncta per cell counts into euploid, trisomies only, monosomies only and all other aneuploid embryos. We performed a Kruskal Wallis test with multiple comparisons. It is worth noticing that the difference between euploid and monosomies only (and those that contained both) was statistically significant, while the difference between euploid vs trisomies only and trisomies only vs monosomies only was not statistically significant. These differences contradict the studies on monosomic cell lines that found that proteotoxic stress and autophagy are not present and specific to trisomic cell lines. Here we also decided to keep this specific protein expression analysis out of the manuscript due to the above-mentioned limitations.

**Author response image 2. sa3fig2:** 

(7) Line 329: "a trisomy 12 meiotic chromosomal abnormality in one reversine-treated embryo." What does it mean? Why meiotic chromosomal abnormality when the reversine treatment was administered 4 days after fertilization? In the discussion, the authors state "presumed meiotic," but this should be discussed and described more clearly.

Since reversine induces mitotic abnormalities of different types leading to chromosomally mosaic embryos, we could not identify these induced abnormalities using inferCNV on the RNAseq of TE biopsies of said embryos. However, we were not aware of the karyotype of the embryos that were used for these experiments, as they were thawed after they had been cryopreserved at day 3 of development and had not been subjected to genetic testing. This makes it possible that some of those embryos we used for the reversine experiments in fact carried endogenously acquired meiotic and mitotic chromosomal abnormalities. Since we are only able to detect by inferCNV aneuploidies homogeneously present in the majority of the cells of the sequenced biopsy, we only picked up this trisomy 12. It is possible that this was not a meiotic abnormality but a miotic one originating at the first cleavage and present at a high percentage of cells in the blastocyst. At any rate, the exact origin of this aneuploidy has no further implications for the results of the study. We clarified this in the manuscript (lines 310-315).

(8) Line 422: "The gene expression profiles suggest that the accumulation of autophagic proteins in aneuploid embryos is caused by increased autophagic flux due to differential expression of the p53 target gene DNA Damage Regulated Autophagy Modulator-1 (DRAM1), rather than by inhibition of autophagy (Supplementary Table 2)." This is highly speculative, as the authors do not have any evidence to support this statement.

To validate this finding we have now stained 7 euploid and 11 aneuploid embryos with a DRAM1 antibody. We found DRAM1 protein to be significantly enriched in the cytoplasm of TE cells but not in the ICM of aneuploid embryos when comparing with euploid embryos (Fig. 3s,t). This data is consistent with the finding that autophagy is increased in the TE and not the ICM of aneuploid human embryos. (Fig 4l-o). Potential implications of DRAM1 expression have been mentioned in the discussion.

(9) The figure legends are confusing. They are mixed up with the methods and some key information are missing.

We revised all figure legends accordingly and removed the experimental set-up figures from the manuscript to reduce any confusion. The methods section was revised and expanded.

(10) In Figure 1, what is the difference between "activated" and "deregulated"?

Since we analyzed our RNA-seq dataset with the method proposed by reviewer 1 we now generated normalized enrichment scores. The terms activated and deregulated are thus not present anymore.

(11) The p62 images are not really clear. There might be more puncta (not obvious, though), but the staining intensity seems lower in the representative images.

We do not agree with the reviewer that there might be more p62 puncta (purple), however, we agree that it was not clearly visible from the pictures. Below we show an example of the counting mask (in green) of the aneuploid embryo from figure 3i, where one can clearly appreciate that all the puncta are captured by the counting mask. In this case, the software counted 1704 puncta. To further clarify, we now added a zoom of a randomly chose ROI of the p62 staining’s to figure 3i.

**Author response image 3. sa3fig3:** 

(12) The authors claim that there are differences between lineages in response to aneuploidy, such as autophagy not being activated in the OCT4+ lineage, etc. However, the differences are very small and based on a small number of embryos. It is difficult to draw far-reaching conclusions based on a small number of experiments (Fig. 4n-r). The authors also claim in the Abstract that they demonstrated "clear differences with previous findings in the mouse", which are however difficult to identify in the text.

We agree with the reviewer that our conclusions on figures 4l-o were based on a small number of embryos. We have increased as much as possible the sample size. This is challenging due to the constrictions in accessing human embryos, and especially the limited number of embryos with meiotic complex aneuploidy. We have performed immunostainings for LC3B, OCT4 and GATA4 of six additional euploid and four additional aneuploid human embryos. This did not change our overall findings that aneuploid embryos upregulate autophagy in the TE rather than the ICM (Figure 4l-o). After the inclusion of additional embryos, we removed our speculation from the manuscript that autophagy is present in ICM cells of already differentiated cells towards EPI/PrE.

We have rephrased the abstract to state that we highlight a few differences with previous findings in the mouse. Here we focused especially on the different transcriptomic response of reversine treated embryos, that aneuploid mouse embryos do not seem to suffer from lineage segregation errors and that the ICM of aneuploid human embryos lacks apoptosis while aneuploid mouse embryos show elimination from the EPI. Likewise, we highlighted the similar stress responses and that we could give novel insights into p53 mediated autophagy and apoptosis activation through DRAM1 in aneuploid TE cells but not the ICM.

(13) The text needs thorough editing - long sentences, typos, and grammar errors are frequent. Punctuation is largely missing.

We have revised the text.

References

(1) Victor, A. R. *et al.* One hundred mosaic embryos transferred prospectively in a single clinic: exploring when and why they result in healthy pregnancies. *Fertil Steril* 111, 280–293 (2019).

(2) Martin, A. *et al.* Mosaic results after preimplantation genetic testing for aneuploidy may be accompanied by changes in global gene expression. *Front Mol Biosci* 10, 264 (2023).

(3) Martín, Á. *et al.* Trophectoderm cells of human mosaic embryos display increased apoptotic levels and impaired differentiation capacity: a molecular clue regarding their reproductive fate? *Human Reproduction* 39, 709–723 (2024).

(4) Domingo-Muelas, A. *et al.* Human embryo live imaging reveals nuclear DNA shedding during blastocyst expansion and biopsy. *Cell* 186, 3166-3181.e18 (2023).

(5) Loewer, A., Karanam, K., Mock, C. & Lahav, G. The p53 response in single cells is linearly correlated to the number of DNA breaks without a distinct threshold. *BMC Biol* 11, 1–13 (2013).

(6) Kim, H., Watanabe, S., Kitamatsu, M., Watanabe, K. & Ohtsuki, T. Cell cycle dependence of apoptosis photo-triggered using peptide-photosensitizer conjugate. *Scientific Reports 2020 10:1* 10, 1–8 (2020).

(7) Pollak, N. *et al.* Cell cycle progression and transmitotic apoptosis resistance promote escape from extrinsic apoptosis. *J Cell Sci* 134, (2021).

(8) Neufeld, T. P. Autophagy and cell growth--the yin and yang of nutrient responses. *J Cell Sci* 125, 2359–2368 (2012).

(9) Lanneau, D. *et al.* Heat shock proteins: essential proteins for apoptosis regulation. *J Cell Mol Med* 12, 743 (2008).

(10) Gabai, V. L., Mabuchi, K., Mosser, D. D. & Sherman, M. Y. Hsp72 and Stress Kinase cjun N-Terminal Kinase Regulate the Bid-Dependent Pathway in Tumor Necrosis Factor-Induced Apoptosis. *Mol Cell Biol* 22, 3415 (2002).

(11) Sanchez-Vasquez, E., Bronner, M. E. & Zernicka-Goetz, M. HIF1A contributes to the survival of aneuploid and mosaic pre-implantation embryos. *bioRxiv* 2023.09.04.556218 (2023) doi:10.1101/2023.09.04.556218.

(12) Chousal, J. N. *et al.* Molecular profiling of human blastocysts reveals primitive endoderm defects among embryos of decreased implantation potential. *Cell Rep* 43, (2024).

(13) Corujo-Simon, E., Radley, A. H. & Nichols, J. Evidence implicating sequential commitment of the founder lineages in the human blastocyst by order of hypoblast gene activation. *Development (Cambridge)* 150, (2023).

(14) Regin, M. *et al.* Lineage segregation in human pre-implantation embryos is specified by YAP1 and TEAD1. *Human Reproduction* 38, 1484–1498 (2023).

(15) Saiz, N., Williams, K. M., Seshan, V. E. & Hadjantonakis, A. K. Asynchronous fate decisions by single cells collectively ensure consistent lineage composition in the mouse blastocyst. *Nature Communications 2016 7:1* 7, 1–14 (2016).

(16) Plusa, B., Piliszek, A., Frankenberg, S., Artus, J. & Hadjantonakis, A. K. Distinct sequential cell behaviours direct primitive endoderm formation in the mouse blastocyst. *Development* 135, 3081–3091 (2008).

(17) Hashimoto, M. & Sasaki, H. Epiblast Formation by TEAD-YAP-Dependent Expression of Pluripotency Factors and Competitive Elimination of Unspecified Cells. *Dev Cell* 50, 139-154.e5 (2019).

(18) Hardy, K. Apoptosis in the human embryo. *Rev Reprod* 4, 125–134 (1999).

(19) Ramos-Ibeas, P. *et al.* Embryo responses to stress induced by assisted reproductive technologies. *Mol Reprod Dev* 86, 1292–1306 (2019).

(20) Licciardi, F. *et al.* Human blastocysts of normal and abnormal karyotypes display distinct transcriptome profiles. *Sci Rep* 8, 1–9 (2018).

(21) Maxwell, S. M. *et al.* Investigation of Global Gene Expression of Human Blastocysts Diagnosed as Mosaic using Next-generation Sequencing. *Reproductive Sciences* 1–11 (2022) doi:10.1007/s43032-022-00899-x.

(22) Groff, A. F. *et al.* RNA-seq as a tool for evaluating human embryo competence. *Genome Res* 29, 1705–1718 (2019).

(23) Starostik, M. R., Sosin, O. A. & McCoy, R. C. Single-cell analysis of human embryos reveals diverse patterns of aneuploidy and mosaicism. *Genome Res* 30, 814–826 (2020).

(24) Vera-Rodriguez, M., Chavez, S. L., Rubio, C., Pera, R. A. R. & Simon, C. Prediction model for aneuploidy in early human embryo development revealed by single-cell analysis. *Nat Commun* 6, 7601 (2015).

(25) Sanchez-Ribas, I. *et al.* Transcriptomic behavior of genes associated with chromosome 21 aneuploidies in early embryo development. *Fertil Steril* 111, 991-1001.e2 (2019).

(26) Fuchs Weizman, N. *et al.* Towards Improving Embryo Prioritization: Parallel Next Generation Sequencing of DNA and RNA from a Single Trophectoderm Biopsy. *Sci Rep* 9, 1–11 (2019).

(27) Fernandez Gallardo, E. *et al.* A multi-omics genome-and-transcriptome single-cell atlas of human preimplantation embryogenesis reveals the cellular and molecular impact of chromosome instability. *bioRxiv* 2023.03.08.530586 (2023) doi:10.1101/2023.03.08.530586.

(28) Dürrbaum, M. & Storchová, Z. Effects of aneuploidy on gene expression: implications for cancer. *FEBS J* 283, 791–802 (2016).

(29) Singla, S., Iwamoto-Stohl, L. K., Zhu, M. & Zernicka-Goetz, M. Autophagy-mediated apoptosis eliminates aneuploid cells in a mouse model of chromosome mosaicism. *Nat Commun* 11, 1–15 (2020).

(30) Bolton, H. *et al.* Mouse model of chromosome mosaicism reveals lineage-specific depletion of aneuploid cells and normal developmental potential. *Nat Commun* 7, 1– 12 (2016).

(31) Ohashi, A. *et al.* Aneuploidy generates proteotoxic stress and DNA damage concurrently with p53-mediated post-mitotic apoptosis in SAC-impaired cells. *Nat Commun* 6, 1–16 (2015).

(32) Santaguida, S. & Amon, A. Short- and long-term effects of chromosome missegregation and aneuploidy. *Nature Reviews Molecular Cell Biology* vol. 16 473–485 Preprint at https://doi.org/10.1038/nrm4025 (2015).

(33) Santaguida, S., Vasile, E., White, E. & Amon, A. Aneuploidy-induced cellular stresses limit autophagic degradation. *Genes Dev* 29, 2010–2021 (2015).

(34) Chunduri, N. K. & Storchová, Z. The diverse consequences of aneuploidy. *Nature Cell Biology 2019 21:1* 21, 54–62 (2019).

(35) Dürrbaum, M. *et al.* Unique features of the transcriptional response to model aneuploidy in human cells. *BMC Genomics* 15, 139 (2014).

(36) Pan, J.-A., Ullman, E., Dou, Z. & Zong, W.-X. Inhibition of protein degradation induces apoptosis through a microtubule-associated protein 1 light chain 3-mediated activation of caspase-8 at intracellular membranes. *Mol Cell Biol* 31, 3158–70 (2011).

(37) Stingele, S. *et al.* Global analysis of genome, transcriptome and proteome reveals the response to aneuploidy in human cells. *Mol Syst Biol* 8, 608 (2012).

(38) Tang, Y.-C., Williams, B. R., Siegel, J. J. & Amon, A. Identification of aneuploidyselective antiproliferation compounds. *Cell* 144, 499–512 (2011).

(39) Janssen, A., Van Der Burg, M., Szuhai, K., Kops, G. J. P. L. & Medema, R. H. Chromosome segregation errors as a cause of DNA damage and structural chromosome aberrations. *Science* 333, 1895–1898 (2011).

(40) Li, M. *et al.* The ATM-p53 pathway suppresses aneuploidy-induced tumorigenesis. *Proc Natl Acad Sci U S A* 107, 14188–14193 (2010).

(41) Thompson, S. L. & Compton, D. A. Proliferation of aneuploid human cells is limited by a p53-dependent mechanism. *J Cell Biol* 188, 369–381 (2010).

(42) Yang, M. *et al.* Depletion of aneuploid cells in human embryos and gastruloids. *Nat Cell Biol* 23, 314–321 (2021).

(43) Jones, R. G. *et al.* AMP-activated protein kinase induces a p53-dependent metabolic checkpoint. *Mol Cell* 18, 283–293 (2005).

(44) Chunduri, N. K., Barthel, K. & Storchova, Z. Consequences of Chromosome Loss: Why Do Cells Need Each Chromosome Twice? *Cells 2022, Vol. 11, Page 1530* 11, 1530 (2022).

(45) Krivega, M., Stiefel, C. M. & Storchova, Z. Consequences of chromosome gain: A new view on trisomy syndromes. *American Journal of Human Genetics* vol. 109 2126–2140 Preprint at https://doi.org/10.1016/j.ajhg.2022.10.014 (2022).